# Extending Lagrangian and Hamiltonian Neural Networks with Differentiable Contact Models

**Yaofeng Desmond Zhong**[†],   **Biswadip Dey**[†],   **Amit Chakraborty**[†]

[†]Siemens Technology, Princeton, NJ 08536, USA.
{yaofeng.zhong, biswadip.dey, amit.chakraborty}@siemens.com

## Abstract

The incorporation of appropriate inductive bias plays a critical role in learning dynamics from data. A growing body of work has been exploring ways to enforce energy conservation in the learned dynamics by encoding Lagrangian or Hamiltonian dynamics into the neural network architecture. These existing approaches are based on differential equations, which do not allow discontinuity in the states and thereby limit the class of systems one can learn. However, in reality, most physical systems, such as legged robots and robotic manipulators, involve contacts and collisions, which introduce discontinuities in the states. In this paper, we introduce a differentiable contact model, which can capture contact mechanics: frictionless/frictional, as well as elastic/inelastic. This model can also accommodate inequality constraints, such as limits on the joint angles. The proposed contact model extends the scope of Lagrangian and Hamiltonian neural networks by allowing simultaneous learning of contact and system properties. We demonstrate this framework on a series of challenging 2D and 3D physical systems with different coefficients of restitution and friction. The learned dynamics can be used as a differentiable physics simulator for downstream gradient-based optimization tasks, such as planning and control. [1] [2]

## 1   Introduction

A large class of real-world physical systems evolves in a piecewise-continuous manner. For example, while playing tennis, tennis balls collide with the ground and the rackets with high elasticity but follow smooth trajectories governed in between those collisions. The ability to walk/run depends heavily on the contacts between the legs and the ground. Unfavorable contact properties can significantly hinder this ability; for example, lack of friction makes it very difficult to walk on icy roads. Robotic manipulators and grippers also rely on contacts and collisions to perform their assigned tasks. These examples highlight the importance of contacts and collisions, which can be found everywhere.

Encoding energy conservation into the computation graph of a neural network constitutes an effective way to improve its data-efficiency and generalization performance in inferring the dynamics of a physical system from its trajectory data [1]. However, as these energy-conserving models assume the system trajectories to be smooth and governed by ordinary differential equations (ODE), they cannot model dynamics with contacts and collisions. On the other hand, another line of work, for example, interaction network [2], neural physics engine [3], and iterative neural projection [4] can model collisions and contacts and learn the associated properties. However, they are not ODE-based and hence cannot infer the continuous dynamics governing the smooth portion of the trajectories. A more recent work [5] has encoded a discrete form of the Euler-Lagrange equation while learning

---

[1]Code available at https://github.com/Physics-aware-AI/DiffCoSim.
[2]Video available at https://www.youtube.com/watch?v=DdJ7RLmGOkg.

35th Conference on Neural Information Processing Systems (NeurIPS 2021).

properties of frictionless 2D contacts. Learning of properties associated with frictional contacts and 3D contacts still remains a relatively underexplored topic in the literature.

In this work, we introduce a contact model that can handle frictional contacts both with or without elasticity as well as enforces energy-conservation during the smooth portions of the trajectories. The scope of energy-conserving neural networks are extended by the contact model. The contact model solves convex optimization problems to calculate the jump in velocity during contact. In order to use this contact model in deep learning tasks, we build upon the recent progress on differentiating through convex optimization problems [6]. We demonstrate the performance of the differentiable contact model in learning coefficients of restitution and friction associated with a variety of 2D and 3D contacts. In addition, we also demonstrate the framework as a differentiable physics simulator and test it in downstream planning tasks.

## 1.1 Related Work

**Lagrangian/Hamiltonian-inspired Neural Networks:** In the last few years, an increasing volume of work has proposed neural network models to learn the underlying dynamics from data while enforcing energy conservation. This line of works leverage Lagrangian dynamics [7–12] or Hamiltonian dynamics [13–16] to incorporate the physics prior of energy conservation into deep learning. Recently, Finzi et al. [17] show that using Cartesian coordinates and enforcing explicit constraints improve learning in both Lagrangian and Hamiltonian settings. To learn the underlying dynamics governed by an ODE, many of these prior works have used Neural ODE [18] which can learn an ODE from observed trajectories. However, real systems often exhibit non-smooth trajectories caused by sudden/abrupt changes in the velocity due to contacts and collisions. Although Neural ODE based approaches have recently been extended for learning dynamics with jump discontinuities [19–21], they cannot accommodate the physical constraints (e.g., maximum dissipation principle, non-negative normal force) associated with contacts. Among the energy-conserving neural networks, only [16] attempted to address contacts and collisions; to capture the elastic collision of a billiard ball, it manually reverses the momentum of the ball orthogonal to the contact surface. However, this specialized technique cannot be applied to frictional or inelastic contacts, or objects that can rotate.

**Contact Model:** Our contact model shares similarity with the contact model of MuJoCo [22–24]. However, there are three differences: (1) our contact model handles elastic contacts while MuJoCo only focuses on inelastic contacts; (2) MuJoCo solves the convex optimization problem with a generalization of the Projected Gauss-Seidel method while we leverage the open-source `scs` solver [25] to solve the optimization problem; (3) the dynamics in MuJoCo is described using generalized coordinates, while we use Cartesian coordinates, since it has been shown in previous work [17], the use of Cartesian coordinates improves the learning of system properties. Another category of contact models solve contact impulses by solving a linear complementarity problem (LCP) [26]. Recently, a number of works [27–32] has proposed differentiable LCPs for downstream planning and control tasks. However, their performance on learning the contact properties has yet to be tested.

**Differentiable Simulation:** The recent past has also witnessed a growing interest in differentiable physics simulation that can be used in many downstream tasks (e.g., parameter estimation, planning, and control) [33–38]. Jiang et al. [39] use an LCP formulation to learn contact impulses for perfectly inelastic contacts. DiffTaichi [35] focuses on material point method and only provides intuitively simple contact mechanisms. The support for partially elastic frictional contacts is yet to be provided. Geilinger et al. [36] differentiates through the dynamics solver analytically. Macklin et al. [37] develop a compliant contact model and formulate an implicit time-stepping scheme for integrating the dynamics. Incremental Potential Contact (IPC) [40] uses a custom implicitly time-stepped solver to solve nonlinear intersection-free and inversion-free elastodynamics. However, it is not clear if unknown dynamics and contact properties can be learned using the implicit time-stepping scheme proposed in [37, 40]. NeuralSim [38] formulates a nonlinear complementarity problem to learn contact impulses and then solves it using Projected Gauss-Siedel. GradSim [41] uses a relaxed Coulomb model to learn contacts from video sequences. Le Lidec et al. [42] propose a differentiable physics simulator that can handle conic friction and elasticity. They demonstrate its ability for system identification on a simple 2D system. Chen et al. [43] propose neural event functions to handle instantaneous changes in a continuous system and test it on frictionless bouncing balls. However, these prior differentiable simulation models do not focus on the energy aspect of the system and their performance on the prediction of system energy (of learned dynamics) has not been evaluated.

## 1.2 Contribution

The main contribution of this work is three-fold. First, by introducing a differentiable contact model, we extend the scope of Lagrangian/Hamiltonian-inspired deep learning methods from collision-free systems to more realistic systems with contact and collisions. Second, we demonstrate the simultaneous learning of system and contact properties in a variety of physical systems by integrating the contact model with Constrained Lagrangian/Hamiltonian neural networks (CLNN/CHNN). We show that the learned contact properties, i.e., coefficients of restitution and friction, are interpretable and match the ground truth with high accuracy. Finally, the learned contact model with CLNN/CHNN can be used to solve downstream gradient-based optimization tasks.

## 2 Preliminaries

### 2.1 Rigid body dynamics without contacts

Consider a rigid body system whose configuration at time $t$ is described by a set of coordinates $\mathbf{x}(t) := (x_1(t), x_2(t), ..., x_D(t))$. Then the time evolution of this system can be expressed as the following second-order ODE,

$$\ddot{\mathbf{x}} = \mathbf{h}(\mathbf{x}, \dot{\mathbf{x}}; \mathbf{p}_s), \tag{1}$$

where $\mathbf{p}_s$ denote system properties, which may include inertia of objects $\mathbf{M}(\mathbf{x})$ and potential energy $\mathbf{V}(\mathbf{x})$. The vector-valued function $\mathbf{h}$ can be derived from the laws of physics, e.g, Lagrangian/Hamiltonian dynamics. By introducing $\mathbf{v} := \dot{\mathbf{x}}$, Eqn. (1) can be written as the following first-order ODE

$$\begin{pmatrix} \dot{\mathbf{x}} \\ \dot{\mathbf{v}} \end{pmatrix} = \begin{pmatrix} \mathbf{v} \\ \mathbf{h}(\mathbf{x}, \mathbf{v}; \mathbf{p}_s) \end{pmatrix} = \mathbf{g}(\mathbf{x}, \mathbf{v}; \mathbf{p}_s). \tag{2}$$

There are two popular choices for the coordinates $\mathbf{x}$ – the generalized coordinates and the Cartesian coordinates. The generalized coordinates are usually chosen as a set of independent coordinates which implicitly enforces holonomic constraints (equality constraints, see Appendix B for details) in the system. The Cartesian coordinates are in general not independent of each other, so that the holonomic constraints in the system must be enforced explicitly in the dynamics (2). Although $\mathbf{g}$ is usually derived with generalized coordinates, this work uses Cartesian coordinates and explicit constraints [17] to demonstrate the results. The proposed contact model is independent of the choice of coordinates and $\mathbf{g}$. We provide the expression of $\mathbf{g}$ used in this work in Appendix B.

### 2.2 Rigid body dynamics with contacts

In robotics tasks, the above assumption of no collision and contact no longer holds. For example, legged robots move around through repeated collisions/contacts between the robot legs and the ground, and robot arms grasp objects by making frictional contact with them. The difficulty of modeling these phenomena is that they essentially make the dynamics discontinuous. For example, when a ball hits the ground, its velocity changes from pointing downward to pointing upward in an infinitesimally small period of time, which can be modeled as an instantaneous change in velocity $\Delta v$. In general, contacts, collisions, and joint limits can all be modeled in this way. Algorithm 1 shows the general procedure of modeling rigid body with contacts, where a jump in velocity is calculated by a contact model whenever there exist active contacts.

---

**Algorithm 1:** Rigid Body Dynamics with Contact

**Input :**
| | |
|---|---|
| $t_0, t_1, ..., t_N$ | Sequence of time points |
| $(\mathbf{x}_0, \mathbf{v}_0)$ | Initial condition at $t_0$ |
| $\mathbf{p}_s = (\mathbf{M}(\mathbf{x}), V(\mathbf{x}))$ | System properties |
| $\mathbf{p}_c = (\boldsymbol{\mu}, \mathbf{e}_P)$ | Contact properties |
| $\mathbf{g}(\mathbf{x}, \mathbf{v}; \mathbf{p}_s)$ | First-order system dynamics |

Initialize output trajectories $\mathcal{T} = \{(\mathbf{x}_0, \mathbf{v}_0)\}$.
**for** $i = 0 \rightarrow N - 1$ **do**
  $(\mathbf{x}_{i+1}, \mathbf{v}_{i+1}) \leftarrow$
    `ODESolve`$(\mathbf{g}, (\mathbf{x}_i, \mathbf{v}_i), t_i, t_{i+1})$
  Get active contacts $\mathbf{c}_a$ (collision detection)
  **if** *exist active contacts* **then**
    $\Delta \mathbf{v} \leftarrow$
      `ContactModel`$(\mathbf{x}_{i+1}, \mathbf{v}_{i+1}, \mathbf{c}_a, \mathbf{p}_s, \mathbf{p}_c)$
    $\mathbf{v}_{i+1} \leftarrow \mathbf{v}_{i+1} + \Delta \mathbf{v}$
  $\mathcal{T} \leftarrow \mathcal{T} \cup \{(\mathbf{x}_{i+1}, \mathbf{v}_{i+1})\}$

---

From a simulation perspective, with known system properties, contact properties (coefficients of friction and restitution), and vector field $\mathbf{g}$, the trajectory of the system can be simulated by Algorithm 1. From a learning perspective, we frame the problem as learning unknown system and contact

properties from a given set of trajectories given a model prior of vector field $\mathbf{g}$. In this case, we can parametrize the unknown system and contact properties $(\mathbf{p}_s, \mathbf{p}_c)$ by neural networks and learnable parameters, predict trajectories by Algorithm 1 and minimize the difference between the predicted and actual trajectories by backpropagation. This training scheme requires all operations in the forward pass (Algorithm 1) to be differentiable. There are two key parts in the forward pass – the ODE solver module and the contact model. Operations in the ODE solver are in general differentiable, and Neural ODE [18, 44] provides a framework of backpropagating through ODE solvers with constant memory cost. In this work, we provide a differentiable contact model so that we can extend these previous works to learn rigid body dynamics with contacts.

## 3 A differentiable contact model

In this section, we introduce a differentiable contact model for learning rigid body dynamics with contacts. The proposed contact model solves post-contact velocities by solving contact impulses in two phases [45] – a *compression phase*, starting from the first contact of objects till the maximum compression, and a *restitution phase*, starting right after the compression phase till the separation of objects. We start by presenting the constraints imposed by frictional contacts.

### 3.1 Contact constraints

This work focuses on two types of contact – frictional contact and limit constraint. For any conceptual frictional contact $i$ in a 3D contact space, the contact impulse $\mathbf{f}_i \in \mathbb{R}^3$ must lie in the friction cone,

$$\mu_i f_{i,n} \geq \sqrt{f_{i,t_1}^2 + f_{i,t_2}^2}, \quad \forall i, \tag{3}$$

where $\mu_i \geq 0$ is the coefficient of friction for conceptual contact $i$. In addition, the normal impulses must be non-negative, since objects can only push but not pull others:

$$f_{i,n} \geq 0, \quad \forall i. \tag{4}$$

For any limit constraint such as limit in joint angle or distance, the contact space is essentially one dimensional and the constraints on contact impulses are only $f_{i,n} \geq 0$. This is mathematically equivalent to setting up a 3D contact space like that in frictional contact and letting $\mu_i = 0$ in Eqn. (3).

### 3.2 Contact model in compression phase

The idea behind solving contact impulses during the compression phase is the maximum dissipation principle [46], which states that the compression impulses should maximize the rate of energy dissipation. Equivalently, the compression impulses are those that minimizes the kinetic energy at the end of the compression phase. This can be described by an optimization problem [4, 46, 22, 23]. We choose the following form, which is similar to the one used in Mujoco [22–24],

$$\underset{\mathbf{f}_C^c}{\text{Minimize}} \; \frac{1}{2}(\mathbf{f}_C^c)^T \mathbf{A} \mathbf{f}_C^c + (\mathbf{f}_C^c)^T \mathbf{v}_C^{c-} \tag{5}$$

$$\text{subject to } (3), (4).$$

where $\mathbf{f}_C^c$ denotes the impulses in compression phase, $\mathbf{A}$ is the inverse inertia in the contact space, and $\mathbf{v}_C^{c-}$ represents the velocity in the contact space before the compression phase. This formulation is an approximation to the Signorini condition, please see [47, 24] for more details. A concise derivation of (5) from the first principle is provided in Appendix C.

### 3.3 Contact model in restitution phase

Similarly, we can set up an optimization problem to solve for the contact impulses in the restitution phase $\mathbf{f}_C^r$. We assume the restitution follows Poisson's hypothesis, where the normal components in $\mathbf{f}_C^r$ equals those in $\mathbf{f}_C^c$ scaled by the coefficient of restitution $e_P$. We adopt Poisson's hypothesis instead of the popular Newton's hypothesis used in prior works [2, 27, 35, 16], because the latter might result in unrealistic energy increase in certain systems [48]. Please see Appendix D for a discussion. We set up the following constraint:

$$f_{i,n}^r \geq e_{P,i} \cdot f_{i,n}^c, \quad \forall i. \tag{6}$$

We have inequality instead of equality here since we would like to compensate for existing penetration in the simulation. Since we simulate the rigid body system in discrete time steps, almost every time when a collision is detected, penetration has already occurred among the objects involved in that collision. Consider the case where the collision is perfectly inelastic, i.e., COR $e_P = 0$, then the true normal impulse during restitution phase would be zero, which fails to fix existing unrealistic penetration. By setting up the constraint as in Eqn. (6), a larger normal impulse is allowed to fix existing penetration. The optimization problem in the restitution phase is

$$\underset{\mathbf{f}_C^r}{\text{Minimize}} \; \frac{1}{2}(\mathbf{f}_C^r)^T \mathbf{A} \mathbf{f}_C^r + (\mathbf{f}_C^r)^T (\mathbf{v}_C^{c+} - \mathbf{v}_C^*) \tag{7}$$

subject to (3), (4), (6).

with $\mathbf{v}_C^{c+}$ as the velocity in contact space after the compression phase and $\mathbf{v}_C^*$ as the target velocity, used for fixing penetration. If there's no penetration, $\mathbf{v}_C^* = \mathbf{0}$. A detailed discussion of penetration compensation can be found in Appendix G. From the principle of maximum dissipation, the equality in Eqn. (6) would hold for the solution when COR is large and penetration is small, thus respecting Poisson's hypothesis when penetration can be fixed naturally.

### 3.4 Differentiability

Solving optimization problems (5) and (7) for contact impulses allow us to calculate instantaneous velocity change to perform simulation (Algorithm 1). Moreover, we would like to back-propagate through the contact model to learn unknown properties. In fact, our proposed contact model is differentiable, thanks to recent progress on differentiable convex optimization layers. Both problems (5) and (7) are convex optimization problems with convex quadratic objectives (we show that $\mathbf{A}$ is positive semi-definite in Appendix E) as well as linear constraints and second-order cone constraints. We can then express our problems using disciplined parametrized programming (DPP) and set up these two problems as differentiable layers using CvxpyLayers [6]. Thus, our model is differentiable and can be used in dynamics and parameter learning as well as downstream tasks.

## 4 Experiments: dynamics and parameter learning

### 4.1 Simulated systems

To evaluate the proposed contact model, we simulate five different systems with contacts (Fig. 1) and propose eight dynamics and parameter learning tasks based on these systems with different contact properties (Table 1). Previous work has studied the *bouncing point masses* (Fig. 1(a)) which is often referred to as billiards or bouncing balls [16, 27, 35]. To make this task more challenging, we let the size of each object be different. We also propose the *bouncing disks* (Fig. 1(b)) where each disk can rotate. The 2-pendulum colliding with the ground has been used to study and analyze contact models for more than three decades [49]. We make this task more challenging by studying a *3-Pendulum colliding with the ground* (Fig. 1(c)). The gyroscope is a 3D system that exhibits complex dynamics. A *gyroscope colliding with a wall* (Fig. 1(d)) is a system where the normal contact impulse does not point towards the center of mass (c.o.m), and Newton's hypothesis might give an unrealistic result with increased energy after collisions [48]. The *rope* (Fig. 1(e)) has also been studied in previous works [2, 4]. Please refer to Appendix H for further details about these systems and the tasks.

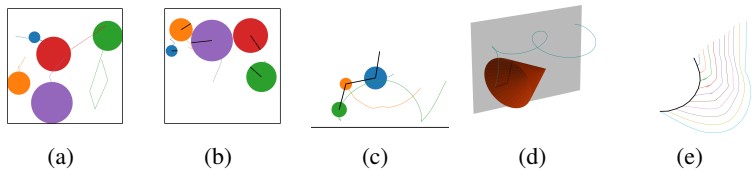

(a)        (b)        (c)        (d)        (e)

Figure 1: Simulated systems with contact. From left to right: (a) bouncing point masses, (b) bouncing disks, (c) chained pendulums with ground, (d) gyroscope with a wall, and (e) rope. The black lines in bouncing disks show the orientations of disks.

Table 1: Benchmark tasks. The columns $D$, $E$, $\max(C)$ denote dimension of the dynamics, number of equality constraints, and the maximum number of contacts that could be simultaneously active , respectively.

| Name | System | $D$ | $E$ | $\max(C)$ | Space | Same $e_P$, $\mu$ for all contacts | Conserve energy |
|---|---|---|---|---|---|---|---|
| BP5-e | Bouncing point masses | 10 | 0 | 8 | 2D | Y | Y |
| BP5 | Bouncing point masses | 10 | 0 | 8 | 2D | N | N |
| CP3-e | Chained pendulums w/ ground | 6 | 3 | 1 | 2D | Y | Y |
| CP3 | Chained pendulums w/ ground | 6 | 3 | 1 | 2D | Y | N |
| BD5 | Bouncing disks | 30 | 15 | 8 | 2D | N | N |
| Rope | Rope | 400 | 0 | $\sim 399$ | 2D | Y | N |
| Gyro-e | Gyroscope w/ a wall | 12 | 7 | 1 | 3D | Y | Y |
| Gyro | Gyroscope w/ a wall | 12 | 7 | 1 | 3D | Y | N |

## 4.2 Dynamics and parameter learning experimental setup

For each simulated system, we jointly learn system and contact properties from trajectory data by extending CLNN/CHNN with the proposed contact model. Fig. 2 shows the architecture.

**Data:** For each task, the training set is generated by randomly sampling 800 collision-free initial conditions and then simulating the dynamics for 100 time steps. Since for some systems, there are very few data points in a trajectory that involves collision, we select a small chunk containing 5 consecutive time steps from each simulated trajectory such that the final training set contains 800 trajectories of length 5, where around half of the trajectories contain collisions and the other half are collision-free. We also make sure that the initial state of these selected chunks is collision-free. The evaluation and test set are generated in a similar way with 100 trajectories, respectively.

**Architecture and training details:** In the experiments, we assume the system properties, i.e., object inertia and potential energy, as well as contact properties, i.e., coefficients of friction and restitution, are unknown and need to be learned from trajectory data. The system properties are parametrized as in CLNN and CHNN [17]. As for contact properties, all coefficient of friction are non-negative, so they are parametrized by scalar learnable parameters passed through *ReLu* function. As each coefficient of restitution lies in the interval of $[0, 1]$, it is parametrized by a learnable parameter passed through a *hard sigmoid* function. The predicted trajectories are generated by running Algorithm 1 with parametrized system and contact properties. We use RK4 as the ODE solver in Neural ODE. We compute the $L_1$-norm of the difference between predicted and true trajectories, and use it as the loss function for training. The gradients are computed by differentiating through Algorithm 1, and learnable parameters are updated using the AdamW optimizer [50, 51] with a learning rate of 0.001.

**Models:** We implement two slightly different versions of the contact model. The first version, referred to as CM, set up optimization problems exactly as stated in (5) and (7). The second version, referred to as CMr, adds a diagonal positive regularization matrix $\mathbf{R} = \epsilon\mathbf{I}$ to $\mathbf{A}$ in (5) and (7), such

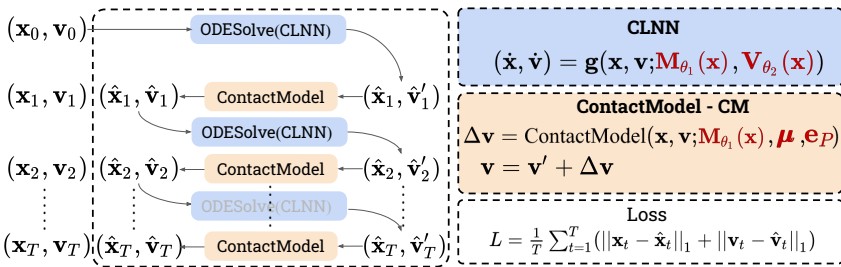

Figure 2: Dynamics and parameter learning schema of the CM-CD-CLNN model. Neural networks and learnable parameters are denoted in red. Predicted trajectories are generated using parametrized CM and CLNN. The difference between the true and predicted trajectories are minimized to learn dynamics and parameters.

that $(\mathbf{A} + \mathbf{R})$ is always positive definite, which ensures a unique global minimum in each problem.[3] These two versions are combined with CLNN and CHNN to set up the following four neural network models: (*i*) CM-CD-CLNN, (*ii*) CM-CD-CHNN, (*iii*) CMr-CD-CLNN, and (*iv*) CMr-CD-CHNN. The "CD" in model names emphasizes that we assume that a collision detection module is given.

**Baselines:** We also set up three baselines. In the first baseline, MLP-CD-CLNN, we calculate the instantaneous velocity change from a multi-layer perceptron (MLP) instead of the proposed contact model. Our second baseline, IN-CP-CLNN, calculates velocity change from an interaction network (IN) [2] without requiring a collision detection module, since IN has the ability to learn collisions and contact. IN requires system and contact properties as input. Here the "CP" in the model name emphasizes true contact properties are given and the system properties learned by CLNN are fed into IN. Our last baseline, IN-CP-SP, is the original interaction network which has shown strong ability in predicting 2D rigid body trajectories without equality constraints, but haven't been tested on systems with equality constraints or 3D systems. The name emphasizes that true system and contact properties are known and are fed into IN. Also, the name indicates no collision detection module is needed in this baseline. To train these baseline models, we transform each trajectory into multiple one-step pairs, as has been done in IN [2]. We also attempted to use the LCP formulation of contact model [27] as a baseline. However, the implementation of gradient computation of the LCP function in [27] results in NaN in our examples. Please refer to Appendix J for additional details. As the forward computation of LCP works as expected, we use LCP-generated training data to test the robustness of our model.

### 4.3 Dynamics and parameter learning results

Our implementation relies on publicly available codebases including Pytorch [52], CHNN [17], Symplectic ODE-Net [14] and Neural ODE [18]. We handle training using Pytorch Lightning [53] for the purpose of reproducibility.

**Prediction:** We report the average relative $L_1$ error over the test trajectories of 7 models on 8 tasks in Fig. 3. In all tasks, our models beat baseline models. The performance difference between CLNN and CHNN is minor since their architectures are similar. In most tasks, CM outperforms CMr. In the BP5-e task, CM beats CMr by 2 orders of magnitude. IN does not perform well even in BP5 tasks since our training set (3.2k one-step pairs) is much smaller than the dataset (1M one-step pairs) used in the IN paper. We also report average relative $L_1$ errors along test trajectories of 50 time steps in Fig. 4, in order to show each model's ability in long term prediction. We observe that our contact models CM and CMr outperform baselines in all tasks.

**Interpretable mass ratio:** Without direct supervision on mass, deep learning algorithms are unlikely to recover the true mass, as pointed out in [14]. However, one can still inspect the ratio of learned mass values to see how well this physical property is learned. The mass ratio plays an important role in determining the motion of objects when they interact with each other, e.g., during collisions. In our BP5 task, CM-CD-CLNN learns the mass ratio $[m_2/m_1, m_3/m_1, m_4/m_1, m_5/m_1] = [2.0001, 6.0036, 8.0014, 10.0024]$, which is very close to the true ratio $[2, 6, 8, 10]$. In fact, our framework is able to accurately learn mass ratios across tasks (please see Appendix I for details).

**Interpretable contact properties:** Table 2 shows the learned contact properties by our 4 models in 6 tasks where the contact properties are the same for all contacts. For all tasks, CM can learn contact

---

[3]The regularization is important for obtaining the inverse dynamics, as stated in [23]. However, the unregularized one learns more accurate dynamics and contact properties, as shown in Fig. 4 and Table 2.

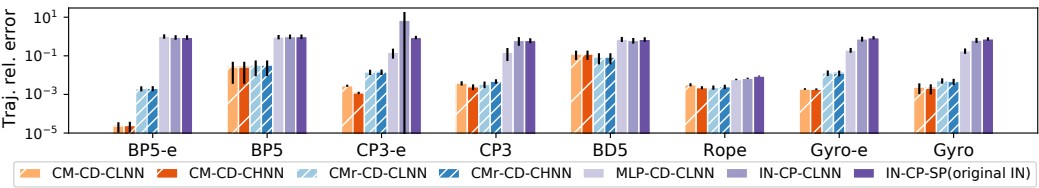

Figure 3: Trajectory relative error (log scale) with 95% confidence interval error bars. Each error is averaged over 100 test trajectories of length 5.

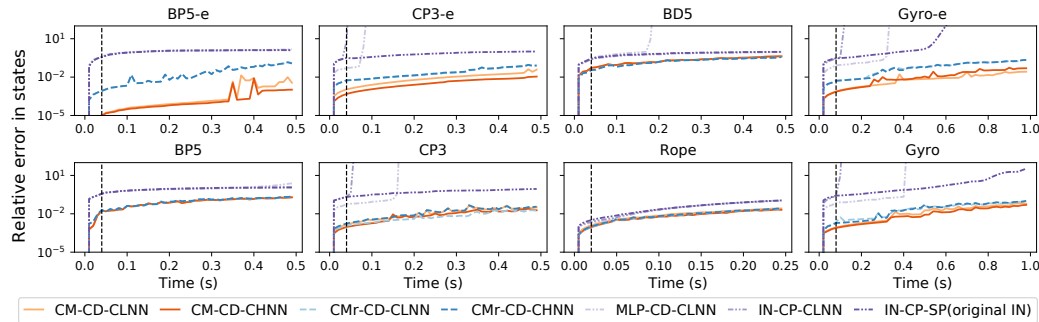

Figure 4: Relative error (log scale) along long test trajectories (50 times steps). Each curve is averaged over 100 test trajectories. Vertical dashed lines show the trajectory length during training.

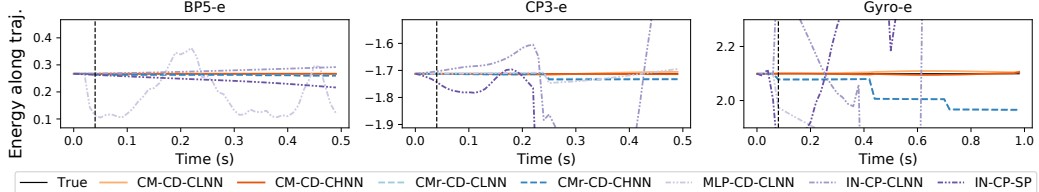

Figure 5: Energy of the predicted trajectories of all 7 models on a sampled test initial condition from BP5-e, CP3-e and Gyro-e tasks. The true energy in each task is represented by the horizontal black line in the middle, which is constant along the trajecotry.

properties accurately which explains its good performance in prediction. CMr is an approximate model and does not infer contact properties as accurately as CM. The interpretability of the learned contact properties along with the mass ratio explains the performance of our framework and shows that the proposed contact model indeed extends Lagrangian and Hamiltonian neural networks.

**Energy:** The prior of Lagrangian/Hamiltonian dynamics conserve energy along each collision-free trajectory, which is one of the reason that Lagrangian/Hamiltonian-based neural network models perform better in prediction and generalization [7, 13, 14, 10]. Fig. 5 illustrates how the total energy changes over time for the predicted trajectories of 7 models on 3 tasks that conserve energy since the contacts are elastic and frictionless (i.e., $e_P = 1, \mu = 0$). Models using CM perform the best in conserving energy in all three tasks since CM learns contact properties perfectly (Table 2). This demonstrates that the proposed contact model can uncover the energy conserving aspect even though energy conservation has not been enforced explicitly. For CP3-e and Gyro-e systems, models using CMr lose energy each time collision happens since they learn positive coefficients of friction in these tasks (Table 2). The baseline models perform the worst in terms of energy conservation.

**Sample efficiency:** We use the BP5 task to demonstrate the sample efficiency of this proposed framework. We vary the training sample size from 25 to 800 trajectories and report the validation loss ($L_1$-norm) of CM-CD-CLNN, MLP-CD-CLNN, and Interaction Network. Figure 6 shows that our framework works well even with limited training data.

Table 2: Learned contact properties from our 4 models on 6 tasks that has unique contact properties for all contacts. Bold numbers are the best learned contact properties in each task across 4 models.

| | BP5-e | | CP3-e | | CP3 | | Rope | | Gyro-e | | Gyro | |
| --- | --- | --- | --- | --- | --- | --- | --- | --- | --- | --- | --- | --- |
| | $\mu$ | $e_P$ | $\mu$ | $e_P$ | $\mu$ | $e_P$ | $\mu$ | $e_P$ | $\mu$ | $e_P$ | $\mu$ | $e_P$ |
| Ground Truth | 0.000 | 1.000 | 0.000 | 1.000 | 0.500 | 0.000 | 0.000 | 0.000 | 0.000 | 1.000 | 0.100 | 0.800 |
| CM-CD-CLNN | **0.000** | **1.000** | **0.000** | **1.000** | 0.500 | 0.005 | 0.026 | 0.000 | **0.000** | **1.000** | **0.100** | **0.800** |
| CM-CD-CHNN | **0.000** | **1.000** | **0.000** | **1.000** | **0.500** | **0.004** | **0.017** | 0.000 | **0.000** | **1.000** | **0.100** | **0.800** |
| CMr-CD-CLNN | **0.000** | **1.000** | 0.002 | 1.000 | 0.500 | 0.023 | 0.037 | 0.011 | 0.002 | 1.000 | 0.099 | 0.892 |
| CMr-CD-CHNN | **0.000** | **1.000** | 0.002 | 1.000 | 0.500 | 0.023 | 0.046 | 0.019 | 0.002 | 1.000 | 0.099 | 0.893 |

**Scalability:** In Table 3, we report the average wall clock time in each iteration (forward pass and backward pass) during training of three sizes of ropes. The time scales approximately linearly with the numbers of coordinates ($D$) and contacts ($C$). See Appendix L for additional results for scalability.

**Robustness:** We evaluate the robustness of our framework by training our model using data generated by LCP formulation and noisy data. (See Appendix K for details.) When trained on LCP data, our framework can learn accurate contact properties in 2D tasks. For the 3D Gyro tasks, the learned contact properties are not as accurate (e.g., learned COR of $0.822$ instead of $0.800$). This is expected since the LCP formulation relaxes the 3D friction cone into a (linear) polyhedral cone and the direction of friction impulses would deviate from those given by our contact model, which is based on the second-order friction cone. In addition, we observe that the performance of our model does not suffer from noisy data since we incorporate

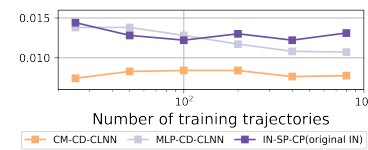

Figure 6: Validation losses for the BP5 task.

Table 3: Average wall clock time in each iteration of ropes. The last column shows increases in time

| $D$ | $\max(C)$ | time (s) | |
|-----|-----------|----------|------|
| 100 | $\sim 99$ | 0.869 | 1x |
| 200 | $\sim 199$ | 1.563 | 1.7x |
| 400 | $\sim 399$ | 3.225 | 3.7x |

strong physics prior into deep learning. For CMr, we also perform an ablation test to investigate the influence of the amount of regularization. By setting the regularizer as $\mathbf{R} = \epsilon\mathbf{I}$, we observe that a smaller $\epsilon$ (e.g. 0.001) result in more accurate learned contact properties. This is expected since a smaller $\epsilon$ approximate (5) and (7) better. However, making $\epsilon$ a learnable parameter does not improve accuracy. Please see Appendix K for more details.

## 5 Experiments: downstream tasks

Since our framework is differentiable, we can use it as a differentiable physics simulator to solve downstream tasks after we have learned the system and contact properties. Here we demonstrate this capability by considering three gradient-based trajectory planning tasks and using CM-CD-CLNN.

**Billiards:** We study the same billiard task as in DiffTaichi [35]. The goal is to find the initial position and velocity of the white ball such that blue ball hit the black target at the 1024th time step. In order to test our framework's ability to solve downstream task and make comparison with DiffTaichi, we assume the parameters such as mass and contact properties are known, the same assumption in DiffTaichi. Fig 7(a) and 7(b) shows the solution found by our proposed model and DiffTaichi, respectively. This task does not have a unique solution since one can place the white ball closer to the billiards with a relatively small initial velocity (e.g. DiffTaichi solution) or place the ball farther away from the billiards with a relatively large initial velocity (e.g. our solution). Fig 7(c) compares the convergence, where the loss is the distance between the black target and the blue ball at the 1024th time step. DiffTaichi has better convergence probably because it implements a simpler contact and dynamics model and it takes time of impact (TOI) into account. The TOI might be able to explain why the optimized positions of the white balls in DiffTaichi and our method are on the right and left of the initial guess, respectively - the gradient w.r.t. the initial position using naive integrator and TOI have different signs (Figure 4 in [35]).

**Throwing:** We present two throwing tasks as shown in Fig. 8. These throwing tasks are simplified versions of similar tasks studied in [36, 37], but we solve the tasks based on learned dynamics while previous works [36, 37] solve them with true dynamics. In the "hit" task (Fig. 8(a)),

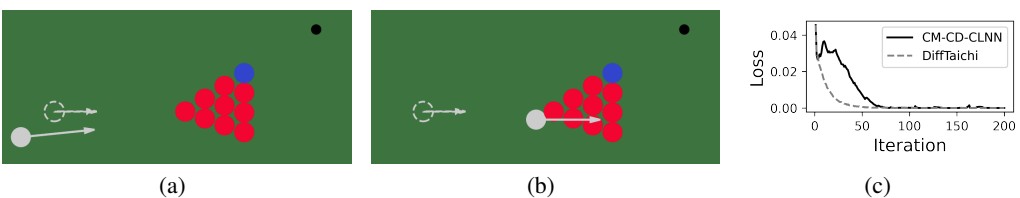

| (a) | (b) | (c) |
|-----|-----|-----|

Figure 7: Billiards. (a), (b): The solid white ball and arrow shows the initial position and velocity optimized by CM-CD-CLNN and DiffTaichi, respectively, while dashed white ball and arrow shows those of the white ball before optimization. (c): Loss as a function of training iterations.

the initial position of the disk is fixed, the goal is to find a desired initial velocity so that the disk reach the target (black circle) after exactly one bounce off the ground. In the "vertical" task (Fig. 8(b)), the initial position and the translational velocity are fixed, so that first half of the center of mass (c.o.m) trajectory (dashed blue) is fixed. The goal is to find a desired initial angular velocity such that the second half of the c.o.m trajectory is as close to a vertical line (dashed black) as possible. In this task we need to learn a counter-clockwise spin such that when the disk bounces off the ground, there are enough friction to stop the horizontal motion.

For these two tasks, we parametrize the initial condition to be learned, simulate the trajectory based on the learned system and contact properties, and minimize the difference between simulated outcome and the goal by gradient descent. We can successfully find the initial conditions to achieve the tasks, evaluated using the true system and contact properties. (Please see the video for additional details).

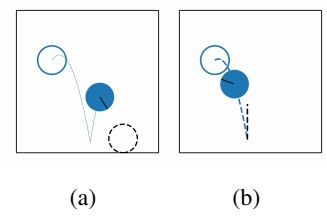

(a)                    (b)

Figure 8: Blue hollow circles indicate the initial position of the disk. (a) the "hit" task. The black hollow circle indicate the target position. (b) the "vertical" task.

## 6   Conclusion

In this work, we have introduced a differentiable contact model, which can capture contact dynamics with different properties. Our contact model extends the applicability of Lagrangian/Hamiltonian-inspired neural networks to enable the learning of hybrid dynamics in rigid body systems and offer interpretability about system and contact properties. We show that the proposed framework achieves better prediction with fewer samples and is robust against noisy data or LCP-generated data. Future works will incorporate model-based control and explore interpretable safe control policies for robotics applications. A particular direction could be to develop appropriate energy shaping control policies [54] and integrate them with this proposed learning framework.

**Limitations:** Our framework assumes a known collision detection module. Although it can be obtained from an idealized touch feedback sensor [5], this information might be unavailable in other scenarios. Future work would explore how to relax this assumption. Our framework might fail to correctly simulate systems which have extremely high mass ratios or stiffness ratios as compared to Macklin et al. [37], where they show their primal method and dual method perform well in high mass ratios and high stiffness ratios scenarios, respectively. Our model might also have challenges in contact-rich systems and might not be as scalable as IPC [40]. Please see Appendix L for additional scalability results. Our framework also uses a mix of acceleration-based simulation (integrating continuous dynamics) and time-stepping methods (calculating instantaneous velocity change) while other simulation methods typically use only one of them. This is because we'd like to use RK4 to better enforce the conservation of energy as done in [14, 17]. However, this choice also makes our simulator not as efficient as other simulators. We would like to compare our method to other differentiable physics model such as NeuralSim [38] and gradSim [41]. However, gradSim has not been open sourced when this work is conducted and it is hard to reproduce model. NeuralSim has its own automatic differentiation engine where gradient are computed one at the time, which is suitable for downstream tasks as demonstrated in [38]; however, it is not suitable for dynamics and parameter learning tasks where a large number of parameters need to be updated based on their gradients. Additional efforts need to made to incorporate NeuralSim with deep learning frameworks. Although we are not able to compare our work with these differentiable physics simulators, these difficulties demonstrate that dynamics and parameter learning with differentiable physics simulators are currently underexplored in the literature.

**Societal impact:** We introduce a framework for data-driven dynamics modelling which uses physics-based priors to improve generalization, sample efficiency, and interpretability. Data-driven dynamics modelling, in general, can have a profound effect in learning-based control synthesis, especially in robotics and automation. However, our proposed framework is still a conceptual proposal and has a very low (around 2) Technology Readiness Level (TRL) [55]. We are yet to fully understand its limitations and failure scenarios that can significantly influence its real-world adoption.

## Acknowledgments and Disclosure of Funding

The authors would like to thank Siemens Technology for supporting this work. Funding in direct support of this work are from Siemens Technology. There are no competing interests.

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
