# Extending Lagrangian and Hamiltonian Neural Networks with Differentiable Contact Models (Supplementary Materials)

**Yaofeng Desmond Zhong**[†], **Biswadip Dey**[†], **Amit Chakraborty**[†]

[†]Siemens Technology, Princeton, NJ 08536, USA.

{yaofeng.zhong, biswadip.dey, amit.chakraborty}@siemens.com

# Appendices

## A    Notation

| | |
|---|---|
| $\mathbf{M}(\mathbf{x})$ | Inertia matrix |
| $V(\mathbf{x})$ | Potential energy |
| $\boldsymbol{\mu}$ | coefficients of friction |
| $\mathbf{e}_P$ | coefficients of restitution |
| $\mathbf{p}_s = (\mathbf{M}(\mathbf{x}), V(\mathbf{x}))$ | System properties |
| $\mathbf{p}_c = (\boldsymbol{\mu}, \mathbf{e}_P)$ | Contact properties |
| $\mathbf{f}_C$ | Contact Impulses |
| $\mathbf{f}_E$ | Equality Constraint Impulses |
| $\mathbf{J}_C(\mathbf{x})$ | Contact Jacobian |
| $\mathbf{J}_E(\mathbf{x})$ | Equality Constraint Jacobian |
| $\mathbf{v}^-/\mathbf{v}^+$ | velocities (in Cartesian space) before/after a general impulse |
| $\mathbf{v}_C^-/\mathbf{v}_C^+$ | velocities (in contact space) before/after a general impulse |
| $\mathbf{v}_C^{c-}/\mathbf{v}_C^{c+}$ | velocities (in contact space) before/after the compression phase |
| $\mathbf{v}_C^{r+}$ | velocities (in contact space) after the restitution phase |

## B    Functional form of the constrained Lagrangian and Hamiltonian dynamics

In this section, we present the functional form of system dynamics that we use in the experiments. Instead of using generalized coordinates, we use Cartesian coordinates. This is because the inertia matrix under Cartesian coordinates are constant and independent of the coordinates, which makes the learning of inertia easier, as pointed out in Finzi et al. [1].

### B.1    Equality constraint Jacobian

Holonomic constraints are equality constraints which can be collected into a column vector $\Phi(\mathbf{x}) \in \mathbb{R}^E$ with equality $\Phi(\mathbf{x}) = \mathbf{0}$. Differentiating this constraint w.r.t. time, we have

$$\dot{\Phi} = (D_\mathbf{x}\Phi)\dot{\mathbf{x}} = (D_\mathbf{x}\Phi)\mathbf{v} = \mathbf{J}_E(\mathbf{x}) \cdot \mathbf{v} = \mathbf{0}, \tag{S.1}$$

35th Conference on Neural Information Processing Systems (NeurIPS 2021).

where we denote the equality constraint Jacobian $\mathbf{J}_E(\mathbf{x}) := D_\mathbf{x}\Phi \in \mathbb{R}^{E \times D}$. Eqn. (S.1) implies that holonomic constraints require the velocity $\mathbf{v}$ to be always in the null space of equality constraint Jacobian $\mathbf{J}_E(\mathbf{x})$. We will use this property to derive impulses caused by equality constraints.

## B.2 Constrained Lagrangian dynamics

The first-order dynamics can be obtained from Finzi et al. [1], which is

$$\begin{pmatrix} \dot{\mathbf{x}} \\ \dot{\mathbf{v}} \end{pmatrix} = \mathbf{g}(\mathbf{x}, \mathbf{v}; \mathbf{p}_s) = \begin{pmatrix} \mathbf{v} \\ \mathbf{M}^{-1}\mathbf{J}_E^T[\mathbf{J}_E\mathbf{M}^{-1}\mathbf{J}_E^T]^{-1}[\mathbf{J}_E\mathbf{M}^{-1}\nabla_\mathbf{x}V - (D_\mathbf{x}(\mathbf{J}_E \cdot \mathbf{v})) \cdot \mathbf{v}] - \mathbf{M}^{-1}\nabla_\mathbf{x}V \end{pmatrix} \tag{S.2}$$

## B.3 Constrained Hamiltonian dynamics

The Hamiltonian dynamics deal with position $\mathbf{x} \in \mathbb{R}^D$ and momentum $\mathbf{p}_\mathbf{x} = \mathbf{M}\mathbf{v}$ instead of $(\mathbf{x}, \mathbf{v})$. The derivation is not as straightforward as in the Lagrangian case. We denote $\mathbf{z} = (\mathbf{x}, \mathbf{p}_\mathbf{x})$. The Hamiltonian equals the total energy of the system and can be written as

$$H(\mathbf{x}, \mathbf{p}_\mathbf{x}) = \frac{1}{2}\mathbf{p}_\mathbf{x}^T\mathbf{M}^{-1}\mathbf{p}_\mathbf{x} + V(\mathbf{x}), \tag{S.3}$$

For the $E$ holonomic constraints $\Phi(\mathbf{x}) = \mathbf{0}$, we can get another $E$ constraints on position and momentum, i.e., $\dot{\Phi}(\mathbf{x}, \mathbf{p}_\mathbf{x}) = 0$, and collect these $2E$ constraints in a vector $\Psi(\mathbf{z}) = (\Phi, \dot{\Phi})$. Then the Hamiltonian dynamics in z can be written as the following differential equations

$$\dot{\mathbf{z}} = \mathbf{J}\nabla_\mathbf{z}H - \mathbf{J}(D_\mathbf{z}\Psi)^T[(D_\mathbf{z}\Psi)\mathbf{J}(D_\mathbf{z}\Psi)^T]^{-1}(D_\mathbf{z}\Psi)\mathbf{J}\nabla_\mathbf{z}H, \tag{S.4}$$

where $\mathbf{J}$ is a symplectic matrix

$$\mathbf{J} = \begin{bmatrix} \mathbf{0} & \mathbf{I}_D \\ -\mathbf{I}_D & \mathbf{0} \end{bmatrix}, \tag{S.5}$$

and $\mathbf{I}_D$ is the $D \times D$ identity matrix. In order to convert the ODE into a set of ODE in $(\mathbf{x}, \mathbf{v})$, we introduce the matrix

$$\tilde{\mathbf{M}}^{-1} = \begin{bmatrix} \mathbf{I}_D & \mathbf{0} \\ \mathbf{0} & \mathbf{M}^{-1} \end{bmatrix}, \tag{S.6}$$

then we obtain the first order ODE

$$\begin{pmatrix} \dot{\mathbf{x}} \\ \dot{\mathbf{v}} \end{pmatrix} = \begin{bmatrix} \mathbf{I}_D & \mathbf{0} \\ \mathbf{0} & \mathbf{M}^{-1} \end{bmatrix}\begin{pmatrix} \dot{\mathbf{x}} \\ \dot{\mathbf{p}}_\mathbf{x} \end{pmatrix} = \tilde{\mathbf{M}}^{-1}\mathbf{J}\nabla_\mathbf{z}H - \tilde{\mathbf{M}}^{-1}\mathbf{J}(D_\mathbf{z}\Psi)^T[(D_\mathbf{z}\Psi)\mathbf{J}(D_\mathbf{z}\Psi)^T]^{-1}(D_\mathbf{z}\Psi)\mathbf{J}\nabla_\mathbf{z}H \tag{S.7}$$

# C  Mathematical Derivation of the differentiable contact model

For simplicity, we present the model by referring to $\mathbf{x}$ and $\mathbf{v}$ as position and velocity in Cartesian space, but the derivation is valid for any other choice of coordinate system. A summary of the notation used here can be found in Section A.

## C.1  Frictional contact and contact Jacobian

Contacts in general can be expressed as inequalities $\Phi_C(\mathbf{x}) \geq \mathbf{0}$. A ball bouncing on the ground, for example, requires the whole ball to be above the ground. When the equality holds for a contact, we refer to the contact as an active contact, otherwise, an inactive contact. If there exists active contacts, contact impulses will cause an instantaneous velocity change. In practice, the set of active contacts is calculated by a collision detection (CD) module.

A conceptual contact can contribute to one or more dimensions in the contact space, corresponding to one or more dimensions of contact impulse. Take Fig. S.1 as an example. Mass 2 at the end of the pendulum would experience a contact impulse $\mathbf{f}_C = (f_n, f_t)$ in the two dimensional contact space - $f_n$ is the component normal to the contact surface, and $f_t$ is the friction impulse tangential to the contact surface. For 3D systems, the contact space is three dimensional with two tangential

components. Assume that the contact space for all active contacts is $C$ dimensional, then we define contact Jacobian $\mathbf{J}_C(\mathbf{x}) \in \mathbb{R}^{C \times D}$, which maps velocities $\mathbf{v}$ in the coordinate space to $\mathbf{v}_C$ in the contact space,

$$\mathbf{v}_C = \mathbf{J}_C(\mathbf{x}) \cdot \mathbf{v}. \tag{S.8}$$

## C.2   Project velocity change into contact space

When there are active contacts, we construct the contact Jacobian $\mathbf{J}_C$ for active contacts. For brevity of notation, we drop explicit dependence on $\mathbf{x}$ from now onward. From Newton's second law, the change of momentum during contact equals the impulses, which can be described as

$$\mathbf{M}\mathbf{v}^+ = \mathbf{M}\mathbf{v}^- + \mathbf{J}_C^T \mathbf{f}_C + \mathbf{J}_E^T \mathbf{f}_E, \tag{S.9}$$

where $\mathbf{v}^-$ and $\mathbf{v}^+$ denote the Cartesian space velocity before and after the instantaneous velocity change, $\mathbf{M}$ is the inertia matrix, $\mathbf{J}_C^T$ maps contact impulses in the contact space $\mathbf{f}_C$ to contact impulses in Cartesian space and $\mathbf{J}_E^T$ maps equality constraint impulses $\mathbf{f}_E$ to equality constraint impulses in Cartesian space. The impulses $\mathbf{f}_C$ and $\mathbf{f}_E$ should not be confused with forces. An impulse is an integral of force over time, which contributes to the change in momentum.

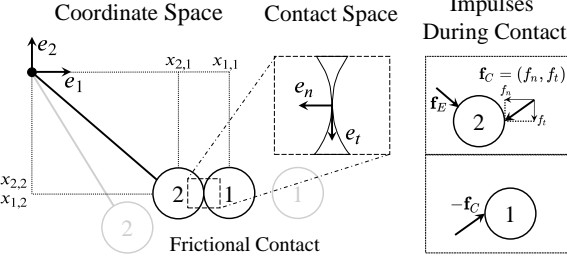

Figure S.1: A ball collide with a pendulum. The equality constraint impulse $\mathbf{f}_E$ ensures that equality constraints are always satisfied.

The equality constraint impulses $\mathbf{f}_E$ is caused by contact impulse $\mathbf{f}_C$. See Fig. S.1 for an intuitive example. Their dependence can be revealed from the fact that the velocity in Cartesian space at any time is in the null space of $\mathbf{J}_E$ (Sec. B.1.) We can left multiply the above equation by $\mathbf{J}_E \mathbf{M}^{-1}$ and solve for $\mathbf{f}_E$,

$$\mathbf{f}_E = -(\mathbf{J}_E \mathbf{M}^{-1} \mathbf{J}_E^T)^{-1} \mathbf{J}_E \mathbf{M}^{-1} \mathbf{J}_C^T \mathbf{f}_C. \tag{S.10}$$

Thus, from Eqn. (S.9) and (S.10), we can express instantaneous velocity change as

$$\mathbf{v}^+ = \mathbf{v}^- + \hat{\mathbf{M}}^{-1} \mathbf{J}_C^T \mathbf{f}_C, \tag{S.11}$$

where

$$\hat{\mathbf{M}}^{-1} = \mathbf{M}^{-1} - \mathbf{M}^{-1} \mathbf{J}_E^T (\mathbf{J}_E \mathbf{M}^{-1} \mathbf{J}_E^T)^{-1} \mathbf{J}_E \mathbf{M}^{-1}. \tag{S.12}$$

$\hat{\mathbf{M}}$ can be interpreted as the inertia that incorporates equality constraints.

In order to solve for contact impulses, we left multiply Eqn. (S.11) by $\mathbf{J}_C$ to project the instantaneous velocity change into the contact space:

$$\mathbf{v}_C^+ = \mathbf{v}_C^- + \mathbf{A}\mathbf{f}_C, \tag{S.13}$$

where $\mathbf{A} = \mathbf{J}_C \hat{\mathbf{M}}^{-1} \mathbf{J}_C^T$, which can be interpreted as the inverse inertia in the contact space. Our contact model solves contact impulses in two phases - the compression phase and the restitution phase, both of which can be described by Eqn. (S.13). Here we express the instantaneous velocity change in two phases as follows

$$\mathbf{v}_C^{c+} = \mathbf{v}_C^{c-} + \mathbf{A}\mathbf{f}_C^c, \tag{S.14}$$

$$\mathbf{v}_C^{r+} = \mathbf{v}_C^{c+} - \mathbf{v}_C^* + \mathbf{A}\mathbf{f}_C^r, \tag{S.15}$$

where $\mathbf{f}_C^c$ and $\mathbf{f}_C^r$ are the contact impulses during the compression phase and the restitution phase and need to be solved by the contact model. The target velocity $\mathbf{v}_C^*$ is included in the restitution phase to compensate existing penetration in the simulation. See Appendix G for details on compensating penetration.

### C.3 Contact model in compression phase

From the maximum dissipation principle, the objective is to minimize the kinetic energy, which leads to the following optimization problem[1]

$$\underset{\mathbf{f}_C^c, \mathbf{v}_C^{c+}}{\text{Minimize}} \frac{1}{2}(\mathbf{v}_C^{c+})^T \mathbf{A}^{-1} \mathbf{v}_C^{c+} \tag{S.16}$$

$$\text{subject to } (S.14), (3), (4).$$

By substitute Eqn. (S.14) into (S.16), we get the optimization problem (5) in the paper. Similarly, optimization problem (7) can be derived.

## D  Elasticity and coefficient of restitution

The elasticity of a collision can be captured by the coefficient of restitution (COR). According to Newton's hypothesis [2], COR is defined as the ratio of the normal relative velocity after the collision to that before the collision, ranging from 0 to 1. This definition of COR can cause unrealistic energy increases when the contact is frictional and the COR is close to 1 [3, 4]. Alternatively, Poisson [5] divides the collision into two phases. The former, referred to as the compression phase, start with the first contact of the bodies and stops at the greatest compression. The latter, referred to as the restitution phase, start right after the compression phase till the separation of bodies. According to Poisson's hypothesis, the COR is defined as the ratio of the normal contact impulse in the restitution phase to that in the compression phase. Poisson's hypothesis is favored in simulation since it will not lead to unrealistic energy increase. For a detailed comparison of different hypotheses, please refer to [6, 7]. In this paper, we define COR $e_P$ in accordance with Poisson's hypothesis.

## E  Proof of positive semi-definiteness of A

By definition, we have $\mathbf{A} = \mathbf{J}_C \hat{\mathbf{M}}^{-1} \mathbf{J}_C^T \in \mathbb{R}^{C \times C}$, where

$$\hat{\mathbf{M}}^{-1} = \mathbf{M}^{-1} - \mathbf{M}^{-1} \mathbf{J}_E^T (\mathbf{J}_E \mathbf{M}^{-1} \mathbf{J}_E^T)^{-1} \mathbf{J}_E \mathbf{M}^{-1}. \tag{S.17}$$

For any real physical system, the inertia matrix $\mathbf{M}$ is symmetric and positive definite. Thus, its inverse exists and can be decomposed using Cholesky decomposition $\mathbf{M} = \mathbf{L}\mathbf{L}^T$. We can then express the inverse inertia that incorporates equality constraints as $\hat{\mathbf{M}}^{-1} = \mathbf{L}(\mathbf{I} - \mathbf{P})\mathbf{L}^T$, where $\mathbf{P}$ is a projection matrix

$$\mathbf{P} = \mathbf{L}^T \mathbf{J}_E^T (\mathbf{J}_E \mathbf{M}^{-1} \mathbf{J}_E^T)^{-1} \mathbf{J}_E \mathbf{L}, \tag{S.18}$$

which satisfies $\mathbf{P}^2 = \mathbf{P}$. A property of projection matrices is that the eigenvalues can only take two values: 1 or 0. By eigen-decomposition, $\mathbf{P}$ and $\mathbf{I} - \mathbf{P}$ can be written as

$$\mathbf{P} = \begin{pmatrix} \mathbf{V}_0 & \mathbf{V}_1 \end{pmatrix} \begin{pmatrix} \mathbf{0} & \mathbf{0} \\ \mathbf{0} & \mathbf{I} \end{pmatrix} \begin{pmatrix} \mathbf{V}_0^T \\ \mathbf{V}_1^T \end{pmatrix} = \mathbf{V}_1 \mathbf{V}_1^T, \tag{S.19}$$

$$\mathbf{I} - \mathbf{P} = \begin{pmatrix} \mathbf{V}_0 & \mathbf{V}_1 \end{pmatrix} \begin{pmatrix} \mathbf{I} & \mathbf{0} \\ \mathbf{0} & \mathbf{I} \end{pmatrix} \begin{pmatrix} \mathbf{V}_0^T \\ \mathbf{V}_1^T \end{pmatrix} - \begin{pmatrix} \mathbf{V}_0 & \mathbf{V}_1 \end{pmatrix} \begin{pmatrix} \mathbf{0} & \mathbf{0} \\ \mathbf{0} & \mathbf{I} \end{pmatrix} \begin{pmatrix} \mathbf{V}_0^T \\ \mathbf{V}_1^T \end{pmatrix} = \mathbf{V}_0 \mathbf{V}_0^T, \tag{S.20}$$

where $\mathbf{V}_0 \in \mathbb{R}^{D \times (D-E)}$. So we can decompose $\mathbf{A}$ into $\mathbf{A} = \mathbf{A}_d^T \mathbf{A}_d$, where $\mathbf{A}_d = \mathbf{V}_0^T \mathbf{L}^T \mathbf{J}_C^T$. Then for any vector $\mathbf{c} \in \mathbb{R}^C$, we have

$$\mathbf{c}^T \mathbf{A} \mathbf{c} = (\mathbf{A}_d \mathbf{c})^T \mathbf{A}_d \mathbf{c} \geq 0, \tag{S.21}$$

which proves that $\mathbf{A}$ is positive semi-definite. If the system does not have equality constraints, $\mathbf{A}$ has full rank and can be decomposed using Cholesky decomposition.

Note that in this forward pass, we need to use `torch.symeig` on matrix $\mathbf{P}$. However, `torch.symeig` operation does not support backward gradient calculation with non-distinct eigenvalues. In practice, we use the implementation in [8] to calculate the gradient of `torch.symeig` operation.

---

[1]Strictly speaking, this form is not correct because $\mathbf{A}$ is invertible only if there exists no equality constraint in the system. When equality constraints do exist, a pseudo-inverse of $\mathbf{A}$ should be used here, and the form Eqn. (5) can still be derived.

# F  Solving contact impulses using CvxpyLayers

In this section, we show an implementation of setting up the differentiable optimization problem in the compression phase using CvxpyLayers and PyTorch. We then show how we use this implementation in CM and CMr.

```python
import torch
import cvxpy as cp
from cvxpylayers.torch import CvxpyLayer

def solve_compression_impulse(
        A_d: torch.Tensor, # shape (D-E, C) or (C, C), decomposition
    of matrix A
        v_: torch.Tensor, # shape (C, 1), velocity before impulse in
    contact space
        mu: torch.Tensor, # shape (n_cld, 1), coefficient of friction
        n_cld: int, # number of active (conceptual) contacts
        d: int, # dimension of each conceptual contact space, can take
     value 2 or 3
    ):
    C = v_.shape[0] # C = n_cld*d
    f = cp.Variable((C, 1)) # impulse variable to be solved
    A_d_p = cp.Parameter(A_d.shape)
    v_p = cp.Parameter((C, 1))
    mu_p = cp.Parameter((mu.shape[0], 1))
    # set up objective, constraints, cvx problem and cvxpylayer
    objective = cp.Minimize(0.5 * cp.sum_squares(A_d_p @ f) + cp.sum(
    cp.multiply(f, v_p)))
    constraints = [cp.SOC(cp.multiply(mu_p[i], f[i*d]), f[i*d+1:i*d+d
    ])
                      for i in range(n_cld)] \
                  + [f[i*d] >= 0 for i in range(n_cld)]
    problem = cp.Problem(objective, constraints)
    cvxpylayer = CvxpyLayer(problem, parameters=[A_d_p, v_p, mu_p],
    variables=[f])
    # forward pass
    impulse, = cvxpylayer(A_d, v_, mu)
    return impulse
```

Listing S.1: Implementation of solving compression phase impulse using CvxpyLayers

```python
...
# get A_d as stated in Section E.
A_d   # shape (D-E, C)
impulse = solve_compression_impulse(A_d, v_, mu, n_cld, d)
```

Listing S.2: pseudocode of solving compression phase impulse in CM

```python
...
# regularization
R = torch.eye(A.shape[0]).type_as(A)*1e-2 # shape (C, C)
A_d = torch.cholesky(A+R, upper=True) # shape (C, C)
impulse = solve_compression_impulse(A_d, v_, mu, n_cld, d)
```

Listing S.3: pseudocode of solving compression phase impulse in CMr

The only difference between CM and CMr is how we construct the matrix $\mathbf{A}_d$. In CM, $\mathbf{A}_d$ is constructed as described in Section E, while in CMr, $\mathbf{A}_d$ is constructed by adding a regularization and performing Cholesky decomposition.

# G  Penetration compensation

To compensate for an existing penetration during restitution phase in the simulation, we use the target velocity $\mathbf{v}_C^* \in \mathbb{R}^C$ and the optimization problem (7). In this section, we discuss how to calculate the

target velocity $\mathbf{v}_C^*$ so that it does not violate the equality constraints of the system. The calculation of $\mathbf{v}_C^*$ might be nontrivial; however, in the backward pass, the gradients of $\mathbf{v}_C^*$ are not required for learning contact properties. Thus, in practice, we do not calculate the backward gradients for every calculation introduced in this section.

To choose the target velocity $\mathbf{v}_C^*$, we first come up with a desired velocity $\mathbf{v}_C^d \in \mathbb{R}^C$. For each direction normal to contact surfaces, the component in $\mathbf{v}_C^d$ is calculated as the depth of penetration divided by integration time interval. For each tangential dimension, the component in $\mathbf{v}_C^d$ is set to zero. This choice of $\mathbf{v}_C^d$ will fix penetration in the next time step. The downside is that for totally inelastic contacts, in the next few time steps, the bodies in collision might separate (because the relative velocity normal to the contact surface is greater than zero), which make the contact looks like partially elastic. This phenomenon can be avoided by using more than one time step to compensate the penetration, i.e., by setting the components in $\mathbf{v}_C^d$ to be a fraction of the depth of the penetration, as shown in the figure below. The reason that we cannot use $\mathbf{v}_C^d$ as the target velocity is that $\mathbf{v}_C^d$

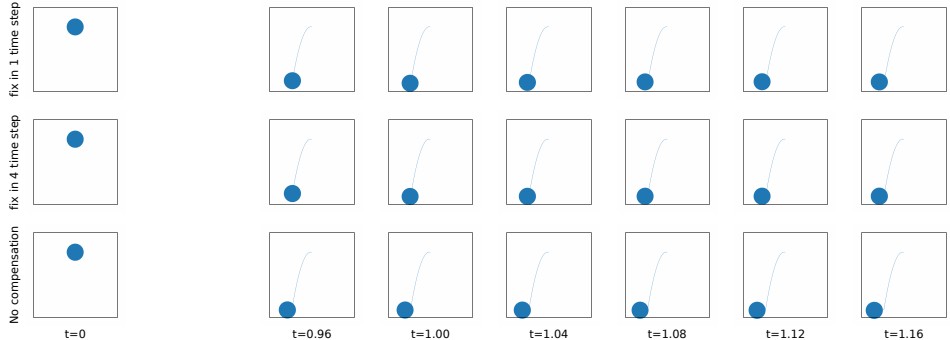

Figure S.2: Different $\mathbf{v}_C^d$ for compensation in a bouncing point mass with gravity and COR=0. **First row**: fixing penetration in 1 time step. The circle bounces off the ground after touching the ground. **Second row**: fixing penetration in 4 time steps. The penetration is fixed and the point mass doesn't bounce up. **Third row**: no penetration compensation. The penetration are not fixed over time.

might violate equality constraints. Take a ball collide with a pendulum as an example (Fig. 1 in main paper), $\mathbf{v}_C^d$ would violate the equality constraint of the pendulum, i.e., the velocity of object 2 can only be perpendicular to the pendulum. Thus, we need to transform $\mathbf{v}_C^d$ into target velocity $\mathbf{v}_C^*$ that satisfies the equality constraints.

The idea of obtaining a target velocity $\mathbf{v}_C^*$ is to project the desired velocity $\mathbf{v}_C^d$ into Cartesian space, make corrections to satisfy equality constraints and then project it back to the contact space. In Section C.2, we showed how to project impulses between contact space and Cartesian space. However, strictly speaking, the projection defined would introduce a scaling if one, say, project a vector from contact space to Cartesian space and back to contact space. This is not a problem for solving contact impulses, but it will be problematic if we have this scaling in calculating target velocity $\mathbf{v}_C^*$. To fix this issue, we need to introduce the pseudoinverse of $\mathbf{J}_C$. Let's assume we have some form of pseudoinverse $\mathbf{J}_C^+$ (we will define it later.) Then the target velocity in Cartesian space is the sum of the desired velocity $\mathbf{v}_C^d$ projected into Cartesian space and a correction term.

$$\mathbf{v}^* = \mathbf{J}_C^+ \mathbf{v}_C^d + \mathbf{J}_E^T \mathbf{v}_E^d \tag{S.22}$$

The equality constraints require $\mathbf{J}_E \mathbf{v}^* = \mathbf{0}$, from which we can solve for $\mathbf{v}_E^d = -(\mathbf{J}_E \mathbf{J}_E^T)^{-1} \mathbf{J}_E \mathbf{J}_C^+ \mathbf{v}_C^d$, then we project the target velocity in Cartesian space $\mathbf{v}^*$ into contact space and get

$$\mathbf{v}_C^* = \mathbf{J}_C \mathbf{v}^* = \mathbf{J}_C (\mathbf{I} - \mathbf{J}_C (\mathbf{J}_E \mathbf{J}_E^T)^{-1} \mathbf{J}_E) \mathbf{J}_C^+ \mathbf{v}_C^d \tag{S.23}$$

The form of the pseudoinverse $\mathbf{J}_C^+$ we use here is dependent on the shape of $\mathbf{J}_C$. We define the pseudoinverse of $\mathbf{J}_C$ as

$$\mathbf{J}_C^+ = \begin{cases} \mathbf{J}_C^T (\mathbf{J}_C \mathbf{J}_C^T)^{-1}, & \text{if } C \leq D \\ (\mathbf{J}_C^T \mathbf{J}_C)^{-1} \mathbf{J}_C^T, & \text{if } C > D \end{cases} \tag{S.24}$$

When $C \leq D$, the dimension of contact space is smaller than that of Cartesian space, we can verify that projecting a velocity from contact space to Cartesian space and back to contact space equals the original velocity, i.e., $\mathbf{J}_C \mathbf{J}_C^+ = \mathbf{I}_C$. When $C > D$, the dimension of contact space is greater than that of Cartesian space, we can verify that projecting a velocity from Cartesian space to contact space and back to Cartesian space equals the original velocity, i.e., $\mathbf{J}_C^+ \mathbf{J}_C = \mathbf{I}_D$.

## H    Simulated systems

**Bouncing point masses.** This system is often referred to as bouncing balls in previous works [9, 10]. We call it bouncing point masses instead since each object is essentially a circle with a point mass at the center and cannot rotate like real balls. For $n$ objects bouncing in the box, there exists $n(n-1)/2$ possible contacts between objects and $4n$ possible contacts between objects and walls. These contacts cannot be all active simultaneously. We set up two tasks of 5 bouncing point masses with different configurations, which will be referred to as BP5-e and BP5. BP5-e is a homogeneous setting, where the masses and radii are the same for all objects, and contact properties ($e_P = 1$ and $\mu = 0$) are the same for all contacts. This task conserves energy since no energy is lost during the collisions and during collision-free periods. BP5 is a heterogeneous setting where the masses and radii are different for different points and contact properties are different for different contacts.

**Bouncing disks.** A real 2D disk has mass spread over the circle and thus can rotate, especially when frictional contacts are involved. Thus we extend the bouncing point masses system to bouncing disks, where all the disks can rotate. We use the extended bodies representation introduced in [1] to embed the motion of disks in Cartesian coordinates. The idea is to use the motion of 3 points - the center of mass as well as the unit vectors aligned with two principle axes - to describe the motion of a disk. Since the relative position of these 3 points are fixed, this representation will introduce 3 equality constraints for each disk. A contact impulse will be distributed properly into these 3 points in a way that obeys the law of physics. Please refer to [1] for more details on this representation. We simulate 5 heterogeneous bouncing disks with heterogeneous contact properties. This task is referred to as BD5.

**Chained pendulums with ground.** The 2-pendulum colliding with ground has been used to study and analyze contact models more than three decades ago [3]. Until recently, some works [1, 11] have studied learning dynamics of N-pendulums without contacts. Here we simulate a 3-pendulum system above the ground where the lowest pendulum can collide with the ground. The masses are located at the joints and the sizes of the joints are different. We follow the convention to assume that pendulums cannot collide with each other. We propose two tasks: CP3-e with $e_P = 1$ and $\mu = 0$, where energy is conserved, and CP3 with $e_P = 0$ and $\mu = 0.5$.

**Gyroscope with a wall.** Gyroscope is a 3D system that exhibits complex dynamics such as precession and nutation. In order to test our contact model in 3D space, we extend the gyroscope system by putting a wall near it so that collisions can happen. The motion of the gyroscope is embedded in Cartesian coordinates using the extended bodies representation[1]. This representation introduces 6 equality constraints. As the gyroscope is attached to a ball joint, one more equality constraint is introduced. We propose two tasks: Gyro-e with $e_P = 1$ and $\mu = 0$, which conserves energy, and Gyro with $e_P = 0.8$ and $\mu = 0.1$.

**Rope.** Our contact model can also capture limits in joint angles and distances, which we show in this rope system. The motion of the rope is described by 10 equally spaced points along the rope. The distance between adjacent points are not fixed as in the chained pendulums system. Instead, the rope can be stretched. The stretch is modelled by elastic springs connecting each pair of adjacent points. We set the maximum stretch and minimum stretch to be 1.2 and 0.8, respectively, which implies that two adjacent points are not allowed to be pushed or pulled by more than $20\%$ of their distance at rest. We also assume that two adjacent segments cannot be bent over a predefined angle (0.2rad). The above stretch and bending constraint can be handled by our contact model with $e = 0$ and $\mu = 0$. During simulation of the rope, a total of 19 "contacts" can be active at the same time, which makes it nontrivial to solve for contact impulses. The force of the spring is modelled via the potential energy, which results in a potential energy function that is not linear in the location of points. This is the only system tested in this work that has a nonlinear potential energy function. This setup is similar to the rope proposed in [12], and differs from string proposed in [9], as the latter impose no bending constraint.

# I Mass ratio details

Here we show the learned mass ratios in BP5-e, BP5, CP3-e and CP3 tasks. We can see the learned mass ratios match the ground truth with high accuracy. This shows our framework learns interpretable mass ratios.

Table S.1: Learned mass ratios in BP5-e

| Mass ratio | $m_2/m_1$ | $m_3/m_1$ | $m_4/m_1$ | $m_5/m_1$ |
|---|---|---|---|---|
| True | 1.0000 | 1.0000 | 1.0000 | 1.0000 |
| CM-CD-CLNN | 1.0000 | 1.0000 | 1.0002 | 1.0003 |
| CM-CD-CHNN | 0.9998 | 1.0000 | 1.0000 | 1.0000 |
| CMr-CD-CLNN | 1.0000 | 0.9993 | 0.9991 | 0.9989 |
| CMr-CD-CHNN | 1.0004 | 0.9994 | 0.9999 | 0.9997 |

Table S.2: Learned mass ratios in BP5

| Mass ratio | $m_2/m_1$ | $m_3/m_1$ | $m_4/m_1$ | $m_5/m_1$ |
|---|---|---|---|---|
| True | 2.0000 | 6.0000 | 8.0000 | 10.0000 |
| CM-CD-CLNN | 2.0000 | 6.0036 | 8.0014 | 10.0024 |
| CM-CD-CHNN | 2.0005 | 6.0020 | 8.0015 | 10.0029 |
| CMr-CD-CLNN | 1.9998 | 6.0004 | 8.0033 | 9.9997 |
| CMr-CD-CHNN | 2.0002 | 6.0001 | 7.9985 | 10.0010 |

Table S.3: Learned mass ratios in CP3 and CP3-e

| Mass ratio | CP3 | | CP3-e | |
|---|---|---|---|---|
| | $m_2/m_1$ | $m_3/m_1$ | $m_2/m_1$ | $m_3/m_1$ |
| True | 0.6500 | 0.7500 | 2.0000 | 1.5000 |
| CM-CD-CLNN | 0.6500 | 0.7502 | 2.0006 | 1.4990 |
| CM-CD-CHNN | 0.6499 | 0.7500 | 1.9996 | 1.4994 |
| CMr-CD-CLNN | 0.6500 | 0.7521 | 2.0002 | 1.5001 |
| CMr-CD-CHNN | 0.6503 | 0.7526 | 2.0009 | 1.5009 |

# J Analysis of LCP baseline

We use the formulation and codebase provided in [13]. The core implementation of differentiable LCP solver is the `LCPFunction` class, which is a subclass of `torch.autograd.Function`. The forward pass of the `LCPFunction` solves a LCP problem and the backward pass computes the gradients. Both the forward pass and the backward pass leverage the primal dual interior point method (pdipm) to compute relevant quantities. However, the provided codebase is outdated and is not compatible with the latest Pytorch release. In order to leverage the codebase to compare it against our method, we first update the core implementation to make it compatible with the latest Pytorch release. We have done a sanity check on the examples provided in the codebase to make sure the updated forward pass and backward pass gives the same results as in the original codebase.

We formulate our 2D and 3D contact problems as LCP problems and use the updated codebase for simulation. As the standard LCP formulation adopts Newton's hypothesis to model elasticity, we adopts Newton's hypothesis in our LCP formulation as well. We plan to use the bouncing point masses system to compare differentiable LCP and our method, since here Newton's hypothesis and Poisson's hypothesis results in the same contact impulses. (In a general system, such as the gyroscope with wall, these two hypotheses result in different contact impulses.) We observe that our LCP formulation generate expected rigid body motions, which shows that the forward pass of `LCPFunction` works well with the CLNN/CHNN dynamics and the extended bodies representation [1]. However, when we try to learn system and contact properties from generated trajectories, we

observe that the backward pass of `LCPFunction` always gives gradients as NaNs. To be specific, the place where NaNs first show up is the `pdipm.solve_kkt()` function call in the backward pass of `LCPFunction`. This indicates a problem with the computation of gradients in the LCP solver. Further investigation is required to see if this is a problem about the primal dual interior point method (pdipm) itself or numerical stability in the implementation.

## K   Robustness analysis details

In this section, we show additional robustness results. These results shows that our model is robust under model mismatch (LCP generated training data) and noise.

Table S.4: Robustness on LCP data (contact properties)

|  | CP3 | | CP3-e | | BP5-e | | Gyro | |
|---|---|---|---|---|---|---|---|---|
|  | $\mu$ | $e_P$ | $\mu$ | $e_P$ | $\mu$ | $e_P$ | $\mu$ | $e_P$ |
| True | 0.500 | 0.000 | 0.000 | 1.000 | 0.000 | 1.000 | 0.100 | 0.800 |
| Trained by CM data | 0.500 | 0.004 | 0.000 | 1.000 | 0.000 | 1.000 | 0.100 | 0.800 |
| Trained by LCP data | 0.500 | 0.005 | 0.003 | 1.000 | 0.000 | 1.000 | 0.100 | 0.822 |

Table S.5: Robustness on LCP data (trajectory relative error w. 95% confidence interval)

|  | CP3 | CP3-e | BP5-e | Gyro |
|---|---|---|---|---|
| Trained by CM data, validated on CM val. data | 2.34e-5(1.29e-5) | 3.85e-3(9.51e-4) | 2.83e-3(3.64e-4) | 2.39e-3(1.36e-3) |
| Trained by LCP data, validated on CM val. data | 2.54e-3(3.73e-3) | 3.73e-3(6.92e-4) | 1.57e-2(5.00e-3) | 5.50e-3(1.67e-3) |
| Trained by LCP data, validated on LCP val. data | 7.35e-4(3.32e-4) | 1.31e-3(3.26e-4) | 5.46e-3(2.30e-3) | 4.51e-4(2.65e-4) |

Table S.6: Robustness on noisy data (contact properties)

| **Noisy Data** | CP3 | | CP3-e | |
|---|---|---|---|---|
|  | $\mu$ | $e_P$ | $\mu$ | $e_P$ |
| True | 0.500 | 0.000 | 0.000 | 1.000 |
| 0 | 0.500 | 0.023 | 0.002 | 1.000 |
| $\mathcal{N}(0, 0.01)$ | 0.496 | 0.036 | 0.004 | 1.000 |
| $\mathcal{N}(0, 0.05)$ | 0.462 | 0.061 | 0.004 | 1.000 |

## L   Scalability details

In this section we show more results on scalability. We use the Rope system to explore how the trajectory relative error changes with the neural network size, number of training trajectory $N$ and the degrees of freedom $D$ (equivalently, the number of contacts). All results reported in the tables are averaged over 100 test trajectories with 95% confidence interval.

Table S.8 shows the effect of network size. Our default network used in CLNN (to approximate potential energy) is an MLP with 3 layers with hidden sizes of 256. We enlarge the network with 6 layers with hidden sizes of 512. We find that large networks doesn't improve our models performance. This is likely because with the default size, the potential energy is already estimated well enough.

Table S.9 shows how the trajectory relative error varies with different number of training trajectories. As expected, for all of the four models, the error decreases with increasing number of training

Table S.7: Robustness on the Regularizer in CMr (contact properties)

| Regularizer Ablation | CP3 | | Gyro | |
|---|---|---|---|---|
| | $\mu$ | $e_P$ | $\mu$ | $e_P$ |
| True | 0.500 | 0.000 | 0.100 | 0.800 |
| $\epsilon = 0.001$ | 0.500 | 0.007 | 0.100 | 0.811 |
| $\epsilon = 0.01$ | 0.500 | 0.023 | 0.099 | 0.892 |
| $\epsilon = 0.1$ | 0.501 | 0.180 | 0.100 | 0.886 |
| learnable | 0.497 | 0.453 | 0.100 | 0.861 |

trajectories. For CM-CD-CLNN, it seems the decreasing trend hasn't converge yet. For MLP-CD-CLNN, the errors doesn't change much from N=800 to N=12800. For IN-CP-CLNN and IN-SP-CP, it is hard to tell if the decrease has converged or not, but it is clear that they have the highest errors across four models (each row). If we compare CM-CD-CLNN and MLP-CD-CLNN, we can clearly see that the gap between our model and the baseline decreases from N=25 to N=800 and increases from N=800 to N=12800. If we compare our model and the other two baselines, the gap decreases from N=25 to N=800, but the trend from N=800 to N=12800 is unclear. The decreasing trend might be unexpected. The underlying reason is that our model performs well with a small amount data. For our model, the difference between N=25 and N=12800 is just 2.9e-4. For baseline models, this difference (between N=25 and N=12800) is at least one order of magnitude higher. Since our model has strong physics priors, this result is expected and shows that our model is data efficient.

Table S.10 shows the trajectory relative error (the same metric used in Figure 3) of ropes discretized in different number of segments (The configurations in Table 3). We find that the performance gap between our model and baselines is the smallest in D=400 scenario. This indicates that our model might not have a clear advantage over baselines in contact-rich scenarios.

These set of results shows that our model might not perform well for contact-rich scenarios. We'd also like to point out that even if our model does not have a clear advantage in contact-rich scenarios, our main contribution is to demonstrate the framework's ability in simultaneously learning of unknown system dynamics and contact properties from trajectory data.

Table S.8: Scalability - large networks

| | CM-CD-CLNN |
|---|---|
| default network | 3.20e-3(5.69e-4) |
| large size | 6.89e-3(5.07e-4) |

Table S.9: Scalability - different training trajectories

| | CM-CD-CLNN | MLP-CD-CLNN | IN-CP-CLNN | IN-SP-CP |
|---|---|---|---|---|
| N=25 | 1.91e-3(3.06e-4) | 2.96e-2(8.04e-3) | 3.73e-2(3.41e-3) | 5.96e-2(7.10e-3) |
| N=50 | 1.87e-3(9.60e-4) | 1.25e-2(2.32e-3) | 3.06e-2(2.35e-3) | 6.49e-2(7.71e-3) |
| N=100 | 2.04e-3(2.91e-4) | 7.36e-3(9.63-e4) | 2.10e-2(1.13e-3) | 2.82e-2(1.92e-3) |
| N=200 | 1.92e-3(3.12e-4) | 8.66e-3(1.04e-3) | 2.68e-2(1.61e-3) | 3.59e-2(2.39e-3) |
| N=400 | 1.90e-3(3.40e-4) | 7.02e-3(7.76e-4) | 1.44e-2(1.01e-3) | 2.71e-2(1.61e-3) |
| N=800 | 3.20e-3(5.69e-4) | 5.98e-3(5.21e-4) | 6.87e-3(5.11e-4) | 8.94e-3(5.30e-4) |
| N=1600 | 2.27e-3(3.51e-4) | 5.94e-3(5.26e-4) | 6.86e-3(4.88e-4) | 9.19e-3(5.27e-4) |
| N=3200 | 1.64e-3(2.73e-4) | 5.98e-3(5.27e-4) | 6.21e-3(5.14e-4) | 7.33e-3(5.26e-4) |
| N=6400 | 1.57e-3(2.27e-4) | 5.92e-3(5.26e-4) | 5.90e-3(5.27e-4) | 7.72e-3(5.85e-4) |
| N=12800 | 1.20e-3(2.88e-4) | 5.93e-3(5.29e-4) | 5.90e-3(5.27e-4) | 7.40e-3(5.31e-4) |

# References

[1] Marc Finzi, Ke Alexander Wang, and Andrew Gordon Wilson. Simplifying Hamiltonian and Lagrangian Neural Networks via Explicit Constraints. volume 33, 2020.

Table S.10: Scalability - different number of contacts

|       | CM-CD-CLNN      | MLP-CD-CLNN     | IN-CP-CLNN      | IN-SP-CP        |
|-------|-----------------|-----------------|-----------------|-----------------|
| D=100 | 1.97e-3(4.72e-4) | 1.85e-2(2.53e-3) | 2.31e-2(2.12e-3) | 2.93e-2(2.05e-3) |
| D=200 | 9.39e-4(3.26e-4) | 1.08e-2(8.91e-4) | 1.27e-2(7.76e-4) | 1.37e-2(8.14e-4) |
| D=400 | 3.20e-3(5.69e-4) | 5.98e-3(5.21e-4) | 6.87e-3(5.11e-4) | 8.94e-3(5.30e-4) |

[2] Isaac Newton. *The Principia: mathematical principles of natural philosophy*. Univ of California Press, 1999.

[3] Thomas R Kane and David A Levinson. *Dynamics, Theory and Applications*. McGraw Hill, 1985.

[4] William James Stronge. Friction in collisions: Resolution of a paradox. *Journal of Applied Physics*, 69(2):610–612, 1991.

[5] Siméon D Poisson. Mechanics, vol. ii. *Trans. HH Harte, Longman, London*, 1817.

[6] Shlomo Djerassi. Collision with friction; Part A: Newton's hypothesis. *Multibody System Dynamics*, 21(1):37, 2009.

[7] Shlomo Djerassi. Collision with friction; Part B: Poisson's and Stornge's hypotheses. *Multibody System Dynamics*, 21(1):55, 2009.

[8] Muhammad Firmansyah Kasim. Derivatives of partial eigendecomposition of a real symmetric matrix for degenerate cases. *arXiv preprint arXiv:2011.04366*, 2020.

[9] Peter Battaglia, Razvan Pascanu, Matthew Lai, Danilo Jimenez Rezende, and Koray Kavukcuoglu. Interaction Networks for Learning about Objects, Relations and Physics. In *Advances in Neural Information Processing Systems*, volume 29, pages 4502–4510, 2016.

[10] Michael B Chang, Tomer Ullman, Antonio Torralba, and Joshua B Tenenbaum. A compositional object-based approach to learning physical dynamics. In *International Conference on Learning Representations*, 2017.

[11] Yaofeng Desmond Zhong, Biswadip Dey, and Amit Chakraborty. Benchmarking Energy-Conserving Neural Networks for Learning Dynamics from Data. *arXiv preprint arXiv:2012.02334*, 2020.

[12] Shuqi Yang, Xingzhe He, and Bo Zhu. Learning physical constraints with neural projections. In *Advances in Neural Information Processing Systems*, volume 33, 2020.

[13] Filipe de Avila Belbute-Peres, Kevin Smith, Kelsey Allen, Joshua B Tenenbaum, and J Zico Kolter. End-to-end differentiable physics for learning and control. In *Advances in neural information processing systems*, volume 31, pages 7178–7189, 2018.