# OpenReview forum: "Extending Lagrangian and Hamiltonian Neural Networks with Differentiable Contact Models"
_NeurIPS.cc/2021/Conference — NeurIPS 2021 Poster_

### Official Review · Reviewer_hJGq · 2021-07-14

**Rating:** 6
**Confidence:** 3

**Summary:**

This paper designs a differentiable contact model and connects it to Lagrangian and Hamiltonian neural networks, enabling simultaneous learning of the contact and system properties. The authors also conduct comprehensive experiments to demonstrate the accuracy, effectiveness, scalability, and robustness of their method.

**Limitations And Societal Impact:**

I did not see any potential negative societal impact of their work. The main limitation of this work from my point of view is its significance, which is described in the last part.

**Main Review:**

Originality: Yes, this is my first time reading a differentiable contact model incorporated into Lagrangian and Hamiltonian neural networks. There have been several differentiable contact models that learn frictional coefficients from data, and the authors have cited them.

Quality:
The authors have conducted comprehensive experiments.

Clarity:
The overall writing is good. I just have some difficulties understanding implementation details.
1. Is there somewhere I can find the structure of the used CLNN/CHNN? Am I right that ODESolve is parameterized by networks while ContactModel is not an NN? Some overview diagrams might help to understand.
2. What is the formulation to compute the relative L1 errors? In figure 3, why are the baseline purple curves approximating 10^0 instead of exploding? It will also be better if standard deviations are shown in the figure as shaded areas.

Significance:
What is the advantage of this method compared to other differentiable physics engines?
1. Others can also learn contact properties. [36] learns contact parameters using neural networks. [29, 30] further use the learned contact properties to perform real-world robot manipulation. Those papers completed similar tasks as this paper does, if not more. The claim in L74 “However, their performance on learning the contact properties has yet to be tested.” might be reconsidered.
2. Moreover, L83 claims “However, these prior differentiable simulation models do not focus on the energy aspect of the system and their performance on the prediction of system energy has not been evaluated”. Are there any experiments to support this argument? As far as I can imagine, unlike dynamics learned by neural networks, differentiable physics engines should naturally preserve the energy, if no collision happens, because they explicitly follow the physical laws. And their energy can be computed analytically.


**Time Spent Reviewing:**

7

---

> ### Author Response · Authors · 2021-08-10
> **Initial response**
>
> - Is there somewhere I can find the structure of the used CLNN/CHNN? Am I right that ODESolve is parameterized by networks while ContactModel is not an NN? Some overview diagrams might help to understand.
>     - If we use Algorithm 1 for simulation, the input g to ODESolve is the ground-truth dynamics. If we use Algorithm 1 for learning, the input g to ODESolve could be CLNN/CHNN. The expression of CLNN/CHNN is given in Appendix B (S.2) (S.7). We have made a diagram, but we find that this rebuttal period is not like the ICLR one where the authors can update manuscripts and supplementary materials. Thus, we cannot update now but we will incorporate it into the camera-ready version. The diagram has been drawn from the perspective of training and hopefully it complements Algorithm 1 to give a high-level picture of our method.
> - What is the formulation to compute the relative L1 errors? In figure 3, why are the baseline purple curves approximating 10^0 instead of exploding? It will also be better if standard deviations are shown in the figure as shaded areas.
>     - The L1 errors are used to set up the loss function for training. The relative trajectory error used in Figure 2 and 3 are formulated is $e_r = ||z(t) - \hat{z}(t)||_2/||z||_2$. We intended to add confidence intervals but as we have seven curves in each plot in Figure 3, adding the confidence interval would make the plots look messy. This is why we only add confidence interval in Figure 2.
> - Others can also learn contact properties. [36] learns contact parameters using neural networks. [29, 30] further use the learned contact properties to perform real-world robot manipulation. Those papers completed similar tasks as this paper does, if not more. The claim in L74 “However, their performance on learning the contact properties has yet to be tested.” might be reconsidered.
>     - NeuralSim [36] might have the ability to learn contact properties, but we didn’t find such experiments in the paper or in the codebase. Since this work is open sourced, we are trying to compare our method to it. We find the biggest challenge is that it has its own way of getting gradients of physical parameters one at a time, which makes it not straightforward to fit into the PyTorch training pipeline. As for [29][30], they did learn the coefficient of friction, but they didn’t show how accurate it is learned. We think the main reason is that these two works focuses more on the downstream tasks and real world examples, e.g., sliding unknown objects. Our work focuses more on broader class of contact models (which includes restitution phase) and evaluates how well we can simultaneously learn dynamics and contact properties. This is why we say “their performance on learning the contact properties has yet to be tested.” We didn’t mean they cannot learn contact properties, it is just the evaluation of how well those properties are learned has not done yet.
> - Moreover, L83 claims “However, these prior differentiable simulation models do not focus on the energy aspect of the system and their performance on the prediction of system energy has not been evaluated”. Are there any experiments to support this argument? As far as I can imagine, unlike dynamics learned by neural networks, differentiable physics engines should naturally preserve the energy, if no collision happens, because they explicitly follow the physical laws. And their energy can be computed analytically.
>     - It is true that differentiable physics engines explicitly follow the physical laws and their energy can be computed analytically, but it doesn’t mean that the energy is naturally preserved. Energy is only conserved if all the forces are conservative forces, e.g. gravity. If we have dissipation in the system, e.g., damped pendulum, or control inputs, e.g., control tasks in OpenAI Gym, the energy will not be conserved. However, we totally understand the reviewer’s confusion. We meant in terms of learning dynamics, the learning based on these prior differentiable simulation does not focus on the energy aspect, while we use CLNN/CHNN, which preserved energy by physics priors. We will make this point clearer.

---

### Official Review · Reviewer_BwHT · 2021-07-16

**Rating:** 4
**Confidence:** 4

**Summary:**

In this work, the authors propose a framework for learning models for dynamical systems that experience contact events by employing a differentiable contact-dynamics model. In simulation, a number of simple experiments are performed to validate such learned models.

**Limitations And Societal Impact:**

Current limitations to this approach include:

Scaling to 3D. Most of the work is in 2D and scaling these approaches to 3D is very difficult. It is unclear whether or not this approach scales to more complex system, particularly in 3D. For example, rotations in 3D will require an orientation representation (e.g., quaternions, MRPs) that is unlikely to be amenable to the lower-level convex optimization problem.

Useful gradients. Because a projection-based cone solver (i.e., SCS) is utilized to solver the contact dynamics problem, the gradients returned that are related to the contact modes are subgradients and potentially do not have information that is useful for finding new contact sequences. This is particularly a concern for locomotion and manipulation tasks that need to explore and discover contact sequences.

Compelling examples. Most of the examples in this work are toy examples compared to the robotic systems utilized in related work which include serial-link manipulators, quadrupeds, and humanoids.

**Main Review:**

The approach utilizes existing tools (i.e., an existing ODE solver and convex solver SCS) to perform time-stepping with collision detection and restitution in forward simulation. The approach is made differentiable with existing neural ODE frameworks and CVXlayers to differentiate through the contact-dynamics optimization problem. Importantly, the contact model is "soft" due to the convex formulation that is regularized.

Originality-
This work is closely related to many other works on differentiable contact simulators. Frankly, it is not nearly as developed as: https://dl.acm.org/doi/pdf/10.1145/3414685.3417766 or http://physbam.stanford.edu/~fedkiw/papers/stanford2019-10.pdf
A strength of this work is utilizing open-source tools, in contrast with tools like MuJoCo.

Quality-
The overall quality of the paper is good. However, it lacks a proper comparison with LCP https://arxiv.org/pdf/2103.16021.pdf or spring-based methods: https://arxiv.org/pdf/2011.04217.pdf. Further, there are no experiments testing the physical validity of the employed soft contact model. For example, validating the model against real-world data or testing control policies in simulation.

Clarity-
The presentation of the work is clear and straightforward. The contact model, specifically, is easily recognizable to an expert in this domain and enables easy analysis of the resulting models.

Significance-
Ultimately, the development of differentiable contact-dynamics simulators has significant potential in the field of robotics and related fields. However, at this stage, the contributions in this paper have not met or advanced existing work.

**Time Spent Reviewing:**

2hrs

---

> ### Author Response · Authors · 2021-08-10
> **Initial response**
>
> - Quality- The overall quality of the paper is good. However, it lacks a proper comparison with LCP https://arxiv.org/pdf/2103.16021.pdf or spring-based methods: https://arxiv.org/pdf/2011.04217.pdf.
>     - Thanks for pointing us to Werling et al. We were not aware of this paper. For NeuralSim [36], we have tried to compare our work to it, but it has its own automatic differentiation implementation, and we find it challenging to integrate it into the PyTorch training pipeline. We are working on it but have not finished the experiments yet.
> - Significance- Ultimately, the development of differentiable contact-dynamics simulators has significant potential in the field of robotics and related fields. However, at this stage, the contributions in this paper have not met or advanced existing work.
>     - By leveraging existing works, we demonstrate the possibility of simultaneous learning of system dynamics and contact properties, which has not been properly evaluated in previous works, even if previous differentiable physics simulators have the ability to obtain gradients w.r.t system parameters.
> - Scaling to 3D. Most of the work is in 2D and scaling these approaches to 3D is very difficult. It is unclear whether or not this approach scales to more complex system, particularly in 3D. For example, rotations in 3D will require an orientation representation (e.g., quaternions, MRPs) that is unlikely to be amenable to the lower-level convex optimization problem.
>     - We’d like to address a confusion here. The reason we leverage CLNN/CHNN [11] is that in the paper they provide an “extended-body representation” where for 3D object, neither quaternions nor Euler angles are needed. Rotations are handled by cartesian coordinates. The paper [11] also shows that Cartesian coordinate data facilitates learning since the potential energy expressed in Cartesian coordinates are usually simpler. We have demonstrated our model in a 3D Gyroscope-with-wall example. It shows that formulating the contacts by convex optimization problem as well as the learning do not have issues.  Whether this model works on more complex 3D systems remains a future work.
> - Useful gradients. Because a projection-based cone solver (i.e., SCS) is utilized to solver the contact dynamics problem, the gradients returned that are related to the contact modes are subgradients and potentially do not have information that is useful for finding new contact sequences. This is particularly a concern for locomotion and manipulation tasks that need to explore and discover contact sequences.
>     - We haven’t tested our model on locomotion and manipulation tasks so we don’t know whether the model would work well in those scenarios. Again, the purpose of this paper is to demonstrate the possibility of learning dynamics and physical properties via differentiable simulation.

---

### Official Review · Reviewer_i24Y · 2021-07-16

**Rating:** 6
**Confidence:** 3

**Summary:**

This paper introduced differentiable contact models into differentiable energy minimizing dynamics formulations like CLNN/CHNN and extends them to handle problem involving collision and contact. An impulse based formulation is used to learn the system and contact properties over a range of tasks.

**Limitations And Societal Impact:**

One limitation is discussed, could me more thorough.

**Main Review:**

Strengths
+ Differentiating through contact is a long standing problem and worth more exploration in the context of machine learning.
+ Results demonstrate a decent step up from the state of art in neural dynamics and differentiable simulation literature.

Weaknesses
- Sec 3 could be grounded with some robot example? Unclear how to extend the theory from 3D points, do you just approximate the robot body with 3D points?

- Approach relies heavily on convex optimization, but most real world problems are non-convex. Where is the assumption reasonable to make and where will this approach fail?

- The impulse formulation is a bit limiting (as it allows one action at the start) from the point of view of robot planning/control that the paper motivates. In practice, it would be more desirable to have the ability to find/execute a sequence of force/torque/velocity based trajectories while accounting for contacts.

- Going beyond dynamics and physical parameter identification, some non planar problem with a 3D robot manipulator focused on the planning/control aspect would add a lot more weight to the experiments. Is there more work to do here on the theoretical side or just a matter of implementation?


**Time Spent Reviewing:**

3

---

> ### Author Response · Authors · 2021-08-10
> **Initial response**
>
> - Sec 3 could be grounded with some robot example? Unclear how to extend the theory from 3D points, do you just approximate the robot body with 3D points?
>     - For our model in its current state, it need to approximate the robot body with 3D points first (if 3D points data is provided).
> - Approach relies heavily on convex optimization, but most real world problems are non-convex. Where is the assumption reasonable to make and where will this approach fail?
>     - Similar to Mujoco simulator, we approximate the LCP problem using a convex optimization problem without explicitly expressing the LCP condition. Modelling contacts with LCP has been used in physics simulators such as Bullet and it is a popular way of modelling frictional contacts. The convex optimization formulation works better than LCP in the sense that it does not approximate the friction cone by polyhedral. The only assumption in our contact model are Columb’s friction law and Poisson’s hypothesis. If the system we try to model has non-negligible static friction, our model might not be as accurate.
> - The impulse formulation is a bit limiting (as it allows one action at the start) from the point of view of robot planning/control that the paper motivates. In practice, it would be more desirable to have the ability to find/execute a sequence of force/torque/velocity based trajectories while accounting for contacts.
>     - We agree that incorporating control would make our work more interesting. We believe similar ideas from Symplectic ODE-Net [4] could be incorporated. That is, simultaneously learn an input matrix (which captures how control interacts with the system). We believe the learning of input matrix should be an easy extension. The hard part is that after learning the dynamics, how to construct a controller. We think this is a bigger topic which should deserve its own contribution. In this work, we focus on demonstrating the possibility of simultaneously learning of system dynamics and contact properties. This is the main reason why we didn’t incorporate control inputs.
> - Going beyond dynamics and physical parameter identification, some non-planar problem with a 3D robot manipulator focused on the planning/control aspect would add a lot more weight to the experiments. Is there more work to do here on the theoretical side or just a matter of implementation?
>     - We believe it is a matter of implementation. In this work, we wrote our own simulator (in Python) with the goal that it can easily fit into the training pipeline of deep learning frameworks, e.g. PyTorch. Of course, our simulator is not as efficient as those implemented in C++, but it is because we couldn’t find an open source differentiable simulator that can be easily integrated with deep learning that we decided to create one ourselves. We do agree that experiments with more complex systems would add value to this work. Due to time constraints, this is what we have at the moment, and we’ll leave more complex systems as future works.

---

> > ### Comment · Reviewer_i24Y · 2021-09-03
> > **response to authors**
> >
> > Thank you for answering my questions. The approach has potential, but the evaluation can be improved with planning/control problems. Having read the reviews and rebuttal, I am inclined to keep my original score.

---

### Official Review · Reviewer_Kv95 · 2021-07-19

**Rating:** 4
**Confidence:** 3

**Summary:**

The paper proposes a differentiable contact model for enabling estimation of object and contact properties from observed trajectories and also for computing actions in downstream tasks.

**Limitations And Societal Impact:**

yes

**Main Review:**

The paper has some major concerns that are listed below:

1. I am unclear on where the contributions lie in the proposed method. The equations for differentiable contacts in Section 3 seems to be from existing work as cited by the authors.

2. The paper does not compare to existing differentiable methods such as DiffTaichi for the estimation of parameters experiment. Specifically, Difftaichi is able to go around the non-differentiability at impact by using time of impact (TOI). Is there some limitation in using TOI? How does the method compare to difftaichi in estimating the parameters from trajectories. I think if this cannot be done by difftaichi (using gradient descent on TOI), then it would be a clear contribution that the paper can highlight.

3. The paper is focusing on ensuring energy conservation during contact simulation but does not show applications that can take advantage of this. E.g., in the billiard simulation, difftaichi regresses much faster than the proposed method.

I think the paper needs to have some clear experiments and comparisons to existing differentiable simulation methods (instead of MLP like baselines) to highlight it's contributions. In its current form, the paper doesn't clearly show why the proposed method is better or required compared to existing methods for contact simulation.

**Time Spent Reviewing:**

5

---

> ### Author Response · Authors · 2021-08-10
> **Initial response**
>
> 1. I am unclear on where the contributions lie in the proposed method. The equations for differentiable contacts in Section 3 seems to be from existing work as cited by the authors.
>     - The contact model is indeed modified from previous works as we have pointed out. The contribution of this work is that it demonstrates the simultaneous learning of dynamics and contact properties of physical systems, which has not been systematically evaluated in precious works, even if previous differential simulators are able to obtain gradients.
> 2. The paper does not compare to existing differentiable methods such as DiffTaichi for the estimation of parameters experiment. Specifically, Difftaichi is able to go around the non-differentiability at impact by using time of impact (TOI). Is there some limitation in using TOI? How does the method compare to difftaichi in estimating the parameters from trajectories. I think if this cannot be done by difftaichi (using gradient descent on TOI), then it would be a clear contribution that the paper can highlight.
>     - First, we’d like to point out that DiffTaichi doesn’t have a full-featured contact model. From the DiffTaichi repo, `billiards.py` doesn’t have friction-related implementation. Although `mass_spring_interactive.py` has a definition of `friction=2.5` in L37, the variable is never used in the file. Thus, of course there’s no learning of coefficient of friction in DiffTaichi since frictional contact is not implemented. We think DiffTaichi should be able to learn coefficient of restitution, but it is not yet evaluated. The TOI is important when we inspect the gradient w.r.t. the input coordinates, as shown in DiffTaichi, but it might not be as important when we inspect the gradient w.r.t. contact properties. We didn’t use TOI in our work since in our work we use RK4 integrator, and it is not clear to us how to properly get TOI with RK4 integrators. DiffTaichi uses Euler integrators where TOI can be obtained by a simple interpolation. From our perspective, DiffTaichi’s main focus is MPM, as it only implements a simple contact mechanism, we think the contribution of our work is different from that of DiffTaichi.
> 3. The paper is focusing on ensuring energy conservation during contact simulation but does not show applications that can take advantage of this. E.g., in the billiard simulation, difftaichi regresses much faster than the proposed method.
>     - We didn’t implement TOI for the reasons discussed above, this could be the reason behind the better performance of DiffTaichi on this task. In fact, this might be able to explain why the optimized positions of the white balls in DiffTaichi and our method are on the right and left of the initial guess, respectively. As you can see from Figure 4 in DiffTaichi [33], the gradient w.r.t the initial position of the ball using naive integrator and TOI have different signs. As we only compute the gradient w.r.t. coordinates (where TOI matters) in Section 5 downstream tasks, we only observed this in Section 5. In fact, the lack of TOI seems to have no impact on the throwing examples, which has also been studied in [34][35] without TOI. As for the experiments in Section 4, since we are not taking gradients w.r.t. the coordinates, TOI doesn’t play an important role, as it can be seen that we can correctly learn the contact properties in Table 2.
> 4. I think the paper needs to have some clear experiments and comparisons to existing differentiable simulation methods (instead of MLP like baselines) to highlight it's contributions. In its current form, the paper doesn't clearly show why the proposed method is better or required compared to existing methods for contact simulation.
>     - Recent differentiable simulator such as gradSim [38], ADD [34] and NeuralSim [36] should give more competitive results than our baselines and we would really love to compare our model against them to understand the pros and cons of these models. However, both gradSim and ADD haven’t released their source code yet, which makes it challenging for us to compare. NeuralSim has made their C++ and python binding code open source. The API makes it easy to get the gradient of loss to each variable one at a time but incorporating it into the pytorch ecosystem is not that straightforward. We are still working to embed NeuralSim in the pytorch training pipeline. This is one of the reason why we create our simulator in Python. It is not optimized for speed but it is easy for people to play around and easy to integrate with deep learning frameworks such as PyTorch.

---

### Official Review · Reviewer_edmT · 2021-07-20

**Rating:** 4
**Confidence:** 5

**Summary:**

This paper introduces a differentiable physics simulator for rigid body dynamics with frictional contact. It combines several existing ideas, including Lagrangian or Hamiltonian neural networks for learning dynamics, the soft contact model of MuJoCo, and differentiable optimization techniques to build a simulator that can be differentiated and trained to match trajectory data. Some model-learning benchmarks are performed against existing simulators and some simple optimal control examples are provided using the differentiable dynamics.

**Limitations And Societal Impact:**

The authors statements on limitations and societal impact are adequate

**Main Review:**

This paper's primary contribution is the combination of several existing ideas into a unified differentiable simulation infrastructure.  Since the algorithmic ideas are, on their own, not novel, I would expect to see their combination demonstrated in some compelling ways in this paper. Since simulating contact dynamics is always fraught with compromise, I feel that any paper that claims to offer better accuracy must deal with real data, rather than synthetic data from another simulator that is inevitably full of artifacts and inaccuracies. The authors primarily compare their method to a standard LCP-based simulator, and even use the LCP method's linearization approximations as a way to explain some of the observed prediction errors in their method. I highly recommend the authors expand their experimental evaluation to include data captured from real physical systems, and then to compare their method against several other state-of-the-art methods on predicting the behavior of those physical systems.

**Time Spent Reviewing:**

1

---

> ### Author Response · Authors · 2021-08-10
> **Initial response**
>
> In this work, we try to answer one question: is it possible to learn the evolution of physical systems (with contacts) in a physics-informed way? This has not been investigated much in previous works. Our approach is to extend CHNN/CLNN [11] to incorporate a differentiable contact model so that we can model and learn a broader class of systems with an end-to-end differentiable simulator. Previous works have studied various differentiable physics simulators, but these works focus mostly on downstream task, e.g., optimize the initial condition for specific tasks, instead of the learning of dynamics and properties. The evaluation of these model on recovering the physical properties are currently missing in the literature. In this work, we show that the simultaneous learning of dynamics and contact properties can be achieved by extending CHNN/CLNN with a optimization-based contact model. Although these tools and ideas already exist in the literature, we leverage them to study a problem that is underexplored in the current literature. We agree that we didn’t test our model on real world data, but this is not the main focus of this work.

---

### Author Response · Authors · 2021-08-10
**Thank you!**

We’d like to thank the anonymous reviewers. We greatly appreciate your feedback and comments! We’ve responded to your comments/questions as replies to the individual comments.

---

### Decision · Program_Chairs · 2021-09-28

**Decision:**

Accept (Poster)

**Comment:**

The core algorithmic ideas in this paper are a combination of existing proposals for differentiable simulation, and hence novelty is not clear. From an application perspective, the paper would strengthen by demonstrations on robotic systems typically used in related work which include serial-link manipulators, quadrupeds, and humanoids, for example. Furthermore, the reviewers expected to see claims of better accuracy with handling contact dynamics validated with real data as opposed to data from other simulators. As such the paper does not clear the bar for NeurIPS acceptance in its current form.


**Consistency Experiment:**

NeurIPS has a long history of experimentation. In 2014, NeurIPS ran an experiment in which 10% of submissions were reviewed by two independent committees to quantify the randomness in the review process. This year, we repeated a variant of this experiment to see how the quality of the review process has changed over time.  This paper was part of the experiment and was therefore assigned to two committees (consisting of reviewers, an Area Chair, and a Senior Area Chair) that reached independent decisions.  If both committees made the same recommendation, this recommendation was followed. If a single committee recommended acceptance, the paper was accepted (with the exception of a few cases in which the other committee identified what we considered a fatal flaw, e.g., an error in a key result).

This copy’s committee reached the following decision: **Reject**

The other committee assigned to the paper recommended **Accept (Poster)**.  You can find the other set of reviews, along with any follow up discussion with the authors here:
https://openreview.net/forum?id=pZQrKCkbas